# Regulatory de novo mutations underlying intellectual disability

Matias G De Vas[1],*, Fanny Boulet[2],*, Shweta S Joshi[1], Myles G Garstang[2,3], Tahir N Khan[2,4] [ID], Goutham Atla[1,5,6] [ID], David Parry[7] [ID], David Moore[8], Inês Cebola[1] [ID], Shuchen Zhang[9] [ID], Wei Cui[9] [ID], Anne K Lampe[8], Wayne W Lam[8], Genomics England Research Consortium[10], Jorge Ferrer[1,5,6] [ID], Madapura M Pradeepa[2,3] [ID], Santosh S Atanur[1,11,12] [ID]

The genetic aetiology of a major fraction of patients with intellectual disability (ID) remains unknown. De novo mutations (DNMs) in protein-coding genes explain up to 40% of cases, but the potential role of regulatory DNMs is still poorly understood. We sequenced 63 whole genomes from 21 ID probands and their unaffected parents. In addition, we analysed 30 previously sequenced genomes from exome-negative ID probands. We found that regulatory DNMs were selectively enriched in fetal brain-specific enhancers as compared with adult brain enhancers. DNM-containing enhancers were associated with genes that show preferential expression in the prefrontal cortex. Furthermore, we identified recurrently mutated enhancer clusters that regulate genes involved in nervous system development (*CSMD1*, *OLFM1*, and *POU3F3*). Most of the DNMs from ID probands showed allele-specific enhancer activity when tested using luciferase assay. Using CRISPR-mediated mutation and editing of epigenomic marks, we show that DNMs at regulatory elements affect the expression of putative target genes. Our results, therefore, provide new evidence to indicate that DNMs in fetal brain-specific enhancers play an essential role in the aetiology of ID.

## Introduction

Intellectual disability (ID) is a neurodevelopmental disorder characterised by limitations in intellectual functioning and adaptive behaviour [1]. The clinical presentation of ID is heterogeneous, often coexisting with congenital malformations or other neurodevelopmental disorders such as epilepsy and autism [1], and the worldwide prevalence is thought to be near 1% [2]. In the past decade, next-generation DNA sequencing has identified a large set of protein-coding genes underlying ID that harbour pathogenic de novo protein-truncating mutations and copy number variants (CNVs) [1, 3]. Nevertheless, despite this recent progress, only up to 40% of the ID cases can be explained by de novo mutations (DNMs) in the protein-coding regions of the genome [4]. DNMs located in non-coding regions of the genome could therefore account for some cases in which no causal pathogenic coding mutation has been identified.

Previous studies have implicated noncoding mutations in long-range *cis*-regulatory elements, also known as transcriptional enhancers, in monogenic developmental disorders, including preaxial polydactyly (*SHH*) [5, 6], Pierre Robin sequence (*SOX9*) [7], congenital heart disease (*TBX5*) [8], and pancreatic agenesis (*PTF1A*) [9]. Systematic analysis of mutations in evolutionarily ultra-conserved noncoding genomic regions has estimated that around 1–3% of patients with developmental disorders but lacking pathogenic coding mutations could carry pathogenic non-coding DNMs in fetal brain *cis*-regulatory elements [10]. Moreover, large-scale whole-genome sequencing (WGS) of patients with autism spectrum disorder (ASD) has demonstrated that DNMs in conserved promoter regions contribute to ASD, although no significant association was found between enhancer mutations and ASD [11].

Despite these precedents, efforts to implicate enhancer mutations in human diseases face numerous challenges. Importantly, it is currently not possible to readily discern functional enhancer mutations from non-functional or neutral variants based on sequence features. This can be partially addressed through experimental analysis of regulatory DNA mutations. Moreover, we still need a complete understanding of which regulatory regions and

---

[1]Section of Genetics and Genomics, Department of Metabolism, Digestion and Reproduction, Imperial College London, London, UK   [2]Blizard Institute, Barts and The London School of Medicine and Dentistry, Queen Mary University of London, London, UK   [3]School of Biological Sciences, University of Essex, Colchester, UK   [4]Department of Biological Sciences, National University of Medical Sciences, Rawalpindi, Pakistan   [5]Regulatory Genomics and Diabetes, Centre for Genomic Regulation, The Barcelona Institute of Science and Technology, Barcelona, Spain   [6]Centro de Investigación Biomédica en Red de Diabetes y Enfermedades Metabólicas Asociadas, Barcelona, Spain   [7]MRC Human Genetics Unit, University of Edinburgh, Edinburgh, UK   [8]South-East Scotland Regional Genetics Service, Western General Hospital, Edinburgh, UK   [9]Institute of Reproductive and Developmental Biology, Faculty of Medicine, Imperial College London, London, UK   [10]Genomics England, London, UK   [11]NIHR Imperial Biomedical Research Centre, ITMAT Data Science Group, Imperial College London, London, UK   [12]Previous Institute: Centre for Genomic and Experimental Medicine, University of Edinburgh, Edinburgh, UK

Correspondence: santosh.atanur@imperial.ac.uk; p.m.madapura@qmul.ac.uk
*Matias G De Vas and Fanny Boulet contributed equally to this work

which subsequences within the regulatory regions are most likely to harbour disruptive mutations. In addition, one of the biggest challenges in interpreting mutations in regulatory regions is correctly associating regulatory elements with the potential target genes. Systematic identification of tissue-specific promoter–enhancer interaction maps would help identification of regulatory regions that are associated with disease-relevant genes.

ID is a severe early-onset neurodevelopmental phenotype; hence, we hypothesised that ID could result from DNMs in enhancers that are specifically active during fetal brain development rather than the adult brain. Furthermore, more than half of the human enhancers have evolved recently; thus evolutionarily not conserved (12), and advanced human cognition has been attributed to fetal brain enhancers that are gained during evolutionary expansion and elaboration of the human cerebral cortex (13). Therefore, we have reasoned that the regulatory sequences critical for intellectual functions may show sequence constraints within human populations regardless of their evolutionary conservation.

In this study, we deployed WGS analysis, integrative genomic and epigenomic studies, together with experimental functional validations to show that DNMs in patients with ID are selectively enriched in fetal brain enhancers. We further show that DNMs map to enhancers that interact with known ID genes, genes that are intolerant to mutations, and genes specifically expressed in the prefrontal cortex. Furthermore, we identify three fetal brain-specific enhancer (FBSE) domains with recurrent DNMs and provide experimental evidence that candidate mutations alter enhancer activity in neuronal cells. Using dual luciferase assay, we show that the majority of enhancer DNMs have allele-specific activity. Furthermore, using CRISPR/Cas9-mediated editing, we show that enhancer DNM results in the down-regulation of target gene expression. Our results provide a new level of evidence that supports the role of DNMs in neurodevelopmental enhancers in the aetiology of ID.

# Results

## WGS and identification of DNMs

We performed WGS of 63 individuals, including 21 probands with severe intellectual disability (ID) and their unaffected parents, at an average genome-wide depth of 37.6X (Table S1). We identified, on average, 4.18 million genomic variants per individual that included 3.37 million single nucleotide variants (SNVs) and 0.81 million short indels (Table S1). We focused our analysis on DNMs, including de novo CNV, as it has been shown that DNMs contribute significantly to neurodevelopmental disorders (3, 4, 14). We identified 1,261 DNMs (de novo SNVs and indels) in 21 trios. An average of 60 high-quality DNMs were identified per proband including 55.2 SNVs and 4.8 indels per proband (Table S2), which was similar to the number of DNMs identified per proband in previous WGS studies on neurodevelopmental disorders (3, 11, 15). We identified three de novo CNVs in our ID probands (Table S3), which is ~0.14 de novo CNVs per proband. The number of de novo CNVs per proband was similar to the expected number of de novo CNVs per individual (16).

## Protein-coding DNMs and CNVs

The role of protein-truncating mutations in ID is well established. Hence, we first looked at DNMs located in protein-coding regions of the genome. A total of 23 DNMs were located in protein-coding regions (an average of 1.1 DNMs per proband). Of the 23 coding mutations, 15 were non-synonymous coding mutations or protein-truncating mutations. In four ID probands (19% of all analysed probands), we identified various potentially pathogenic mutations in the genes KAT6A, TUBA1A, KIF1A, and NRXN1, all of them previously implicated in ID (1, 3). The mutation in KAT6A resulted in a premature stop codon, whereas genes TUBA1A and KIF1A showed non-synonymous coding mutations, which have been reported as likely pathogenic and pathogenic, respectively, in ClinVar (17) (Table S4). One de novo CNV resulted in the partial deletion of NRXN1, a known ID gene. A family with two affected siblings was analysed for the presence of recessive variants. These findings are in agreement with previous reports on the pathogenic role of DNMs in the ID (14). All the coding DNMs were confirmed by Sanger sequencing (data not shown).

## ID DNMs are preferentially located in FBSEs

In our severe ID cohort, we did not identify pathogenic coding DNMs in 17 ID cases (~81%); hence, we decided to investigate potentially pathogenic mutations in disease-relevant enhancer regions. Our cohort size was relatively small, hence to increase the sample size for statistical analysis, we included 30 previously published severe ID samples in which no pathogenic protein-coding DNMs have been found using WGS (3), yielding a total of 47 exome-negative ID cases.

We hypothesised that DNMs in FBSE could perturb the expression levels of genes that are important for brain development, leading in this way to ID. We, therefore, identified 27,420 FBSEs using the data from the Roadmap Epigenomics project (18) (see the Materials and Methods section). The majority (76.52%) of these FBSEs were found to be candidate cis-regulatory elements (ccREs) defined by ENCyclopedia Of DNA Elements (ENCODE) (19), confirming the regulatory role of these enhancers. In addition, we analysed 8,996 human brain-gained enhancers that are active during cerebral corticogenesis (13).

A total of 83 DNMs were located within FBSEs or human gain enhancers (HGE, an average of 1.77 DNMs per proband), which include 82 de novo SNVs and one de novo indel (Table S5). A total of 52 DNMs were located within FBSEs, 30 DNMs were in human gain enhancers, whereas one overlapped with both fetal brain enhancers and a human gain enhancer. First, we investigated whether in our ID cohort (n = 47), DNMs were enriched in the enhancers that are specifically active in the fetal brain or the enhancers that are active in specific subsections of the adult brain. In our ID cohort, DNMs were significantly enriched in FBSEs and human gain enhancers compared with adult brain-specific enhancers (t test $P = 9.12 \times 10^{-7}$; FDR = $3.68 \times 10^{-6}$; Fig 1A and Table S6). Next, we investigated whether the enrichment of DNMs in fetal brain enhancers can also be observed in healthy individuals or if it was specific to the ID cohort. To investigate this, we used DNMs identified in healthy individuals in the Genome of Netherlands (GoNL) (20) study. On the contrary, in healthy individuals (GoNL), DNMs were

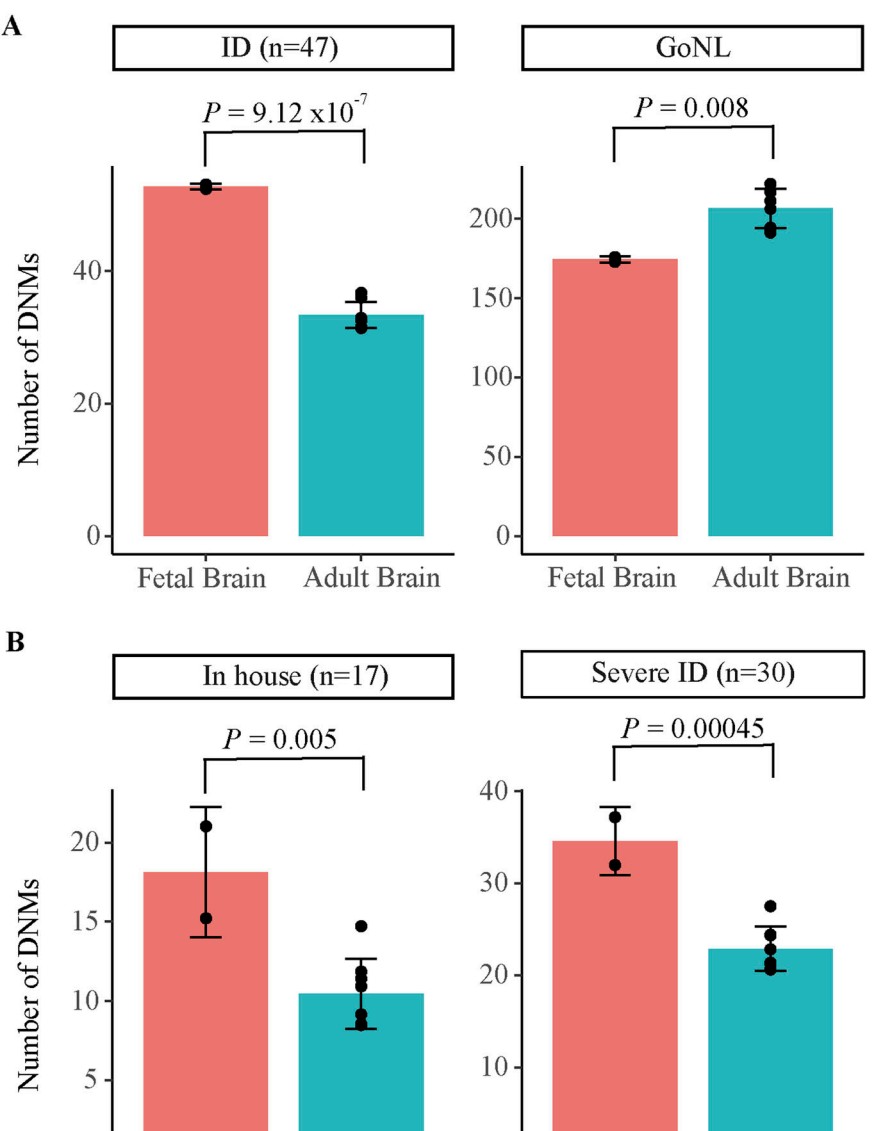

**Figure 1. Enrichment of de novo mutations in fetal brain-specific enhancers.**
**(A)** Enrichment of the observed number of de novo mutations in fetal brain-specific enhancers and human gain enhancers as compared with adult brain subsection-specific enhancers in the intellectual disability cohort (n = 47) and healthy controls (GoNL). **(B)** DNM enrichment in in-house (n = 17) and severe ID cohorts (n = 30). The fetal brain enhancers (orange bars) represent fetal brain-specific and human-gained enhancers. The adult brain enhancers (turquoise bar) represent adult brain subsection-specific enhancers, including angular gyrus, anterior caudate, cingulate gyrus, germinal matrix, hippocampus middle, inferior temporal lobe, dorsolateral prefrontal cortex, and substantia nigra. ID: Intellectual disability, GoNL: Genomes of the Netherland, In house: ID cohort sequenced in this study, Severe ID: Coding-negative ID cohort from reference 3 study.

depleted in FBSEs compared with adult brain-specific enhancers (t test $P = 0.008$; FDR = $8.35 \times 10^{-3}$; Fig 1A and Table S6). Furthermore, as a negative control, we used two tissues (lung and small intestine) that are not related to ID and for which enhancer data were available for both fetal and adult samples in the Roadmap Epigenomics project. We did not observe any enrichment in fetal enhancers as compared with adult enhancers ($P = 0.45$, Table S6) in tissues that are not relevant to ID, suggesting that the DNM enrichment observed in fetal brain enhancers is tissue-specific. When we analysed two ID cohorts, the in-house cohort (n = 17) and severe ID cohort (n = 30), separately, DNMs from both cohorts showed enrichment in FBSEs as compared with adult brain enhancers (t test $P = 0.005$ and $4.5 \times 10^{-4}$; FDR = $6.66 \times 10^{-3}$ and $9 \times 10^{-4}$, respectively, Fig 1B). This suggests that the enrichment of ID DNMs in the fetal brain enhancer was consistent across ID cohorts and results obtained

using a combined dataset (n = 47) were not biased because of one specific ID cohort.

As our method to estimate the enrichment of DNMs in fetal brain enhancers as compared with adult brain enhancers is new, to support our findings, we also performed a two-sample Poisson rate ratio test to estimate enrichment. We found that the DNMs were significantly enriched in the fetal brain enhancers as compared with adult brain enhancers ($P = 0.005$); however, no statistically significant enrichment was observed in fetal lung and fetal small intestine enhancers (negative controls) as compared with adult lung and adult small intestine ($P = 0.58$ and $0.88$, respectively). This suggests that the statistical method used to estimate enrichment is robust and the enrichment of DNMs observed in the fetal brain enhancer is a real reflection of biology. Our results were consistent with the expectation that mutations in enhancers active during

fetal brain development contribute to the aetiology of ID, a severe early-onset neurodevelopmental phenotype, rather than mutations in enhancers active in the adult brain.

### DNM-containing enhancers were associated with ID-relevant genes

We next investigated the hypothesis that DNMs are preferentially located in enhancers connected with genes that are plausible etiological mediators of ID. To identify potential target genes of fetal brain enhancers, we used the following datasets in sequential order: promoter capture Hi-C (PCHi-C) (21) from neuronal progenitor cells (NPC); correlation of H3K27ac ChIP-seq signal at promoters and enhancers across multiple tissues; and promoter–enhancer correlation using chromHMM segmentation data (18). The closest fetal brain-expressed gene was assigned as a target gene for the 24% of the enhancers that remained unassigned after applying these approaches. For all approaches, we restricted our search space to topologically associated domains (TADs) defined in the fetal brain tissue (22) as most enhancer–promoter interactions are restricted by TAD boundaries (23). On average, enhancers were connected to 1.64 genes, whereas each gene was associated with 4.83 enhancers. These findings were consistent with previous reports of enhancer–promoter interactions (24).

Next, we compiled a list of genes that have previously been implicated in ID or related neurodevelopmental disorders, using four gene sets: known ID genes (1, 3), ID genes from Genomics England panel app (https://panelapp.genomicsengland.co.uk/), genes implicated in neurodevelopmental disorders in the Deciphering Developmental Disorder (DDD) project (4), and autism risk genes (SFARI genes) (25). The DNMs predominantly lead to dominant disorders; hence, we selected dominant genes from the four gene lists for enrichment analysis. This resulted in a set of 617 dominant genes previously implicated in neurodevelopmental disorders. The genes associated with the DNM-containing fetal brain enhancers show significant enrichment for known neurodevelopmental disorder genes (17 genes, hypergeometric test $P = 5.4 \times 10^{-5}$; FDR = $1.35 \times 10^{-4}$; Table S7). The most robust enrichment was observed for the DDD genes (15 genes, hypergeometric test $P = 1.05 \times 10^{-5}$; FDR = $5.25 \times 10^{-5}$; Table S7). Out of 47 coding-negative ID patients in 17 patients, DNMs were observed in FBSEs associated with known neurodevelopmental disorder genes.

Next, to gain insights into which biological processes the genes associated with the DNM-containing enhancers are involved in and which tissues they are predominantly expressed, we performed gene ontology enrichment analysis and tissue expression analysis using web-based platform Enrichr (https://maayanlab.cloud/Enrichr). We observed that the target genes of DNM-containing enhancers were not only involved in nervous system development (Gene Ontology—biological process $P = 7.4 \times 10^{-4}$; Table S8), but also predominantly expressed in the prefrontal cortex (ARCHS4 tissue $P = 6.5 \times 10^{-3}$; Table S9), a region of the brain that has been implicated in social and cognitive behaviour, personality expression, and decision-making (26).

The potential functional effect of enhancer mutations is expected to be mediated through the altered expression of target genes. Recently, it has been shown that most known severe haploinsufficient human disease genes are intolerant to loss of

function (LoF) mutations (27). We compared the putative target genes of DNM-containing enhancers with the recently compiled list of genes that are intolerant to LoF mutations (pLI ≥ 0.9) (27). We found that a significantly higher proportion of enhancer DNM target genes were intolerant to LoF mutations than expected (hypergeometric test $P = 4.2 \times 10^{-5}$; Table S10).

Taken together, our analysis shows that DNMs detected in severe ID patients are predominantly found in enhancers that regulate genes that are specifically expressed in the prefrontal cortex, have been previously implicated in ID or related disorders, and exhibit intolerance to LoF mutations.

### ID DNMs in regulatory regions of human brain cell types

The human brain is a most complex tissue composed of multiple cell types and subtypes (28, 29). We found that the genes associated with DNM-containing enhancers were predominantly expressed in the prefrontal cortex. The human cortex undergoes extensive expansion during development (30). The radial glia (RG), cortical stem cells give rise to intermediate progenitor cells (IPCs) and excitatory neurons (eN) which migrate to cortical plate (30, 31), whereas interneurons (iN) migrate tangentially into the dorsal cortex (30, 32). The human brain mainly consists of astrocytes, oligodendrocytes, microglia, neurons, and other cell types (28). It has been shown that the cell type-specific regulatory regions were enriched for the genome-wide association studies risk variants for brain disorders and behavioural traits (28); hence, we investigated regulatory regions in which specific human brain cell types are enriched for DNMs in the ID cohort. We obtained cell type-specific regulatory region annotations for developing human cortex (30) and human prefrontal cortex (28) from previous publications.

Cell type-specific open chromatin (ATAC-seq) data were available for radial glia, IPCs, excitatory neurons, and interneurons from developing cortex (30). We did not find enrichment for ID DNMs in open chromatin regions (ATAC-seq peaks) for any of the developing cortex cell types (Table S11A). However, when the analysis was restricted to interacting open chromatin regions as defined by H3K4me3-mediated PLAC-seq, only interacting open chromatin regions of IPCs showed significant enrichment for ID DNMs compared with the GoNL DNMs after multiple test corrections (Fisher's exact test $P = 5.18 \times 10^{-5}$; FDR = $4.14 \times 10^{-4}$) suggesting that DNMs affecting highly interacting regulatory regions of IPCs might be functional. Interestingly, this signal was driven by DNMs overlapping promoter regions rather than enhancers (Table S11B). Therefore, we performed an enrichment analysis by selecting ATAC-seq peaks that overlap with the promoter regions (±2 kb of TSS) of protein-coding genes. Promoter regions of all four cell types were enriched for ID DNMs as compared with GoNL DNMs.

Of four human brain prefrontal cortex cell types (astrocytes, oligodendrocytes, microglia, and neuronal cells), enhancer regions of only microglia ($P = 0.0073$; FDR = 0.012; Table S11C) and neuronal cells ($P = 0.037$; FDR = 0.049; Table S11C) showed enrichment after multiple test corrections. On the contrary, all four human brain prefrontal cortex cell types showed significant enrichment for ID DNMs compared with GoNL DNMs in promoter regions after correcting for multiple tests. A total of 44 DNMs overlapped with the promoter regions of at least one of the four human brain cell types (Table S11D). Interestingly, the

majority (70%) of the DNMs overlapped with the promoters that were active in all four cell types and 88.63% overlapped with promoters that were active in three or more cell types.

Enrichment of ID DNMs in promoter regions of all the cell types of developing cortex and prefrontal cortex, whereas enrichment of ID DNMs specifically in enhancer regions of IPCs and neuronal cells highlight the role of disease-relevant cell type-specific enhancers in disease aetiology.

### Recurrently mutated enhancer clusters

In our cohort, we did not observe individual enhancers being recurrently mutated (containing two or more DNMs from unrelated probands). It has been shown that enhancers that regulate genes important for tissue-specific functions often cluster together (33). Therefore, we investigated whether clusters of fetal brain enhancers, that is, sets of enhancers associated with the same gene showed recurrent DNMs. We observed that the enhancer clusters associated with *CSMD1*, *OLFM1*, and *POU3F3* were recurrently mutated with two DNMs in each of their enhancer clusters (Fig 2). To test the enrichment of observed number of DNMs in the enhancer clusters associated with *CSMD1*, *OLFM1*, and *POU3F3*, we used a previously defined framework for interpreting DNMs (34). In short, the model determines the mutability of a given base by taking into consideration one nucleotide on each side (trinucleotide context). All three enhancer clusters showed statistically significant

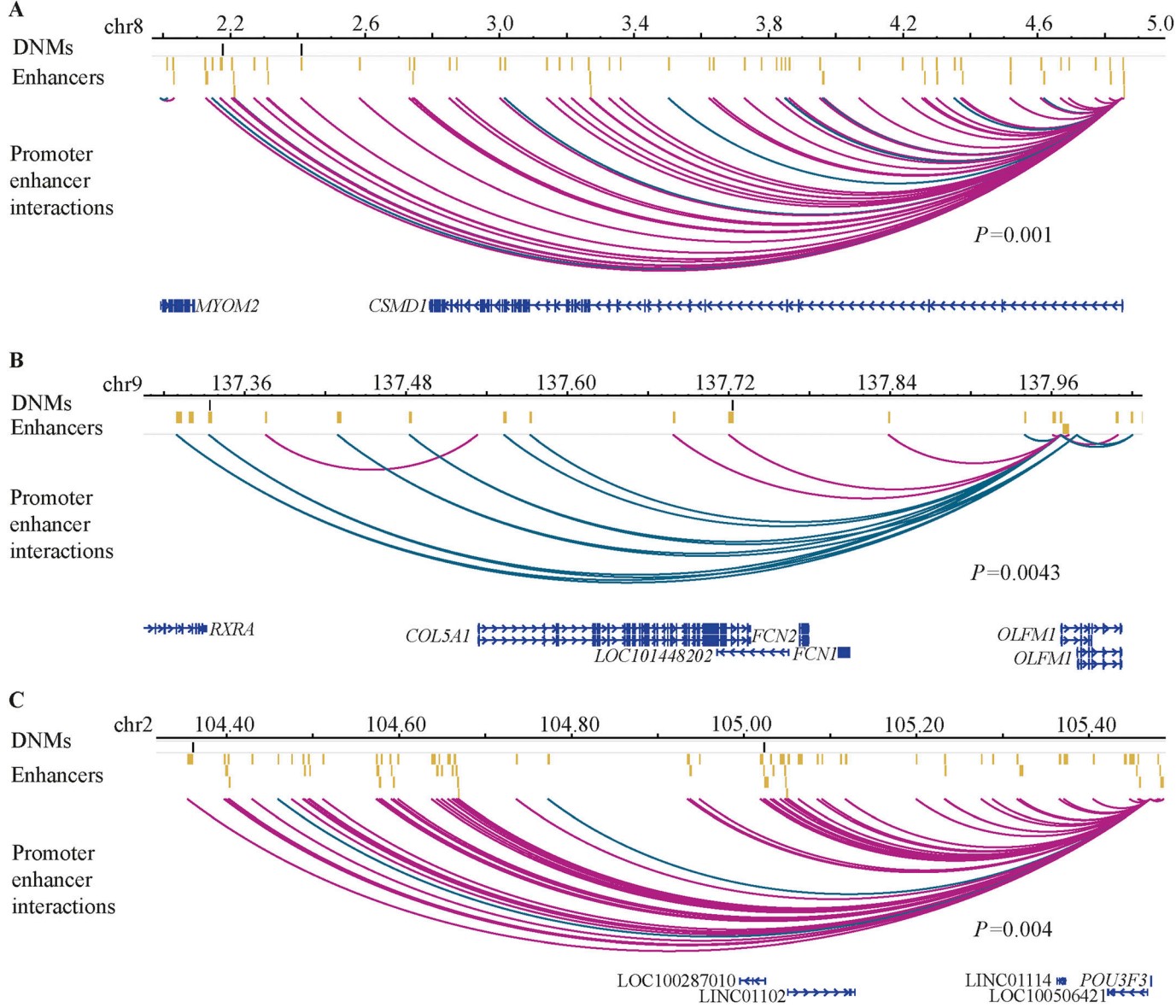

**Figure 2. Enhancer clusters with recurrent DNMs in the ID cohort.**
**(A)** Recurrent DNMs in the *CSMD1* enhancer cluster. **(B)** Recurrent DNMs in the *OLFM1* enhancer cluster. **(C)** Recurrent DNMs in the *POU3F3* enhancer cluster. Black lines indicate DNMs, whereas yellow bars indicate enhancers. Magenta arcs represent fetal brain-specific enhancer–promoter interactions, whereas dark green arcs represent human gain enhancer–promoter interactions. The scales are provided as million base pairs.

enrichment even after multiple test corrections (Poison test $P = 1 \times 10^{-3}$, $4.3 \times 10^{-3}$, $4 \times 10^{-3}$; FDR = $3.1 \times 10^{-3}$, $4.3 \times 10^{-3}$, $4.3 \times 10^{-3}$, respectively). The presence of three enhancer clusters with recurrent DNMs within the cohort of 47 ID probands was significantly higher than expected (permutation test $P = 0.016$). All three genes (*CSMD1*, *OLFM1*, and *POU3F3*) play a role in the nervous system development ([35], [36], [37]). Altogether, the known role of these genes in nervous system development and the presence of recurrent mutations in their enhancer clusters in the ID cohort suggest that these enhancer DNMs may contribute to ID.

### DNMs in *CSMD1*, *OLFM1*, and *POU3F3* enhancer clusters in a large ID cohort

To gain further insights into the enhancer clusters that showed more than one DNM in our cohort, we explored WGS data from large cohorts of neurodevelopmental disorders. Genomics England Limited (GEL) has sequenced 6,514 patients with intellectual disability. In GEL, no pathogenic coding variants were found in 3,169 ID patients for which WGS data of unaffected parents were available (trio WGS). We analysed DNMs from these 3,169 samples to find additional evidence supporting *CSMD1*, *OLFM1*, and *POU3F3* enhancer DNMs. We found three individuals with *CSMD1* enhancer DNMs, five patients with DNMs in the *OLFM1* enhancer cluster, and 15 ID patients with DNMs in the *POU3F3* enhancer cluster. Only the enhancer cluster associated with *POU3F3* showed statistically significant enrichment for DNMs in the GEL cohort as compared with the expected number of DNMs estimated using the previously defined framework for DNMs ([34]) ($P = 1.97 \times 10^{-3}$, FDR = $5.9 \times 10^{-3}$). Next, we extracted human phenotype ontology terms of the patients with DNMs in enhancers of these genes. Of three probands with *CSMD1* enhancer DNMs, two showed delayed speech and language development, delayed motor development, microcephaly, and seizures. All five probands with *OLFM1* enhancer DNMs showed delayed speech and language development, whereas three showed autistic behaviour and delayed motor development. Nine out of 15 probands with *POU3F3* enhancer DNMs showed autistic behaviour, global developmental delay, and delayed speech and language development, whereas eight probands showed delayed motor development. Heterozygous mutations in *POU3F3* protein-coding regions have been recently implicated in ID ([38], [39]). Phenotypes of probands with *POU3F3* enhancer mutations match the reported phenotypes of ID patients with coding *POU3F3* mutations ([39]). The phenotypic similarity between the patients harbouring DNMs in the enhancer of the same gene suggests they might be functional.

### Functional disruption of enhancer function by ID DNMs

Enhancers regulate gene expression through the binding of sequence-specific transcription factors (TFs) at specific recognition sites ([40]). DNMs could elicit phenotypic changes because they alter the sequence of putative TF-binding sites or create putative TF-binding sites (TFBS) that impact target gene expression. We used stringent criteria for TF motif prediction and motif disruption (see the Materials and Methods section). The software used to predict the effect of variants on TF motif (MotifbreakR) works only with SNVs

and not the indels; hence, we investigated only 82 de novo SNVs for their effect on TF motif disruption and excluded one de novo indel from this analysis. Of the 82 de novo SNVs that were located in fetal brain enhancers, 32 (39%) were predicted to alter putative TFBS affinity, either by destroying or creating TFBS (Table S12A). The fetal brain enhancer DNMs from ID probands frequently disturbed the putative binding sites of TFs that were predominantly expressed in neuronal cells (Enrichr; ARCHS4 tissue $P = 0.022$; Table S12B). Our results suggest that the enhancer DNMs from ID probands were more likely to affect the binding sites of neuronal transcription factors and could influence the regulation of genes involved in nervous system development through this mechanism.

Of 17 in-house exome-negative probands, at least one DNM was predicted to alter TFBS affinity in 11 probands. To test the functional impact of regulatory mutations on enhancer activity, we randomly selected one DNM each from each of these 11 ID patients (Table S12A). Altogether, we selected 11 enhancer DNMs (Table S13) and investigated their functional impact in luciferase reporter assays in the neuroblastoma cell line SH-SY5Y. Of the 11 enhancers containing DNMs, 10 showed significantly higher activity than the negative control (empty vector) in at least one allelic version (either WT or mutant allele), indicating that they do indeed function as active enhancers in this neuronal cell line (Fig 3). Amongst these 10 active fetal brain enhancers, nine showed allele-specific activity, with five showing loss of activity and four showing gain of activity of the DNMs (Fig 3). The *CSMD1* enhancer cluster had two DNMs (chr8:g.2177122C>T and chr8:g.2411360T>C) in two unrelated ID probands (Family 6 and Family 3, respectively). Both DNMs yielded a gain of enhancer activity compared with the WT allele (Fig 3). By contrast, two DNMs in the *OLFM1* enhancer cluster (chr9:g.137722838T>G and chr9:g.137333926C>T) from two unrelated ID probands (Family 4 and Family 12, respectively) caused a loss of activity (Fig 3). Furthermore, for the majority of the DNMs (8 out 9), the allele-specific activity was consistent with the predicted effect of the MotifBreakR (Table S13). For example, *CSMD1* enhancer DNMs disrupt the binding site of TCF7L1; a transcriptional repressor and luciferase assay shows that the mutant allele results in a gain of enhancer activity. These results demonstrate that selected DNMs from ID patients in fetal brain enhancers alter predicted TF-binding affinity and have a functional impact on enhancer activity assays.

### *SOX8* enhancer DNM leads to reduced expression of the *SOX8* gene

The luciferase reported assay is an episomal assay; hence, we randomly selected one DNM from the list of DNMs that showed allele-specific activity for investigation in genomic context using CRISPR. The luciferase reporter assays showed that the enhancer DNM (chr18:g.893142:C>A) from a family 14 probands results in a loss of enhancer activity (Fig 3). The promoter capture Hi-C data from neuronal progenitor cells showed a strong interaction between DNM containing the enhancer and the promoter region of *SOX8* (positive strand) and *LMF1* (negative strand) genes (Fig 4A), suggesting that the *SOX8* and/or *LMF1* genes could be regulated by this enhancer in neuronal cells. The DNM-containing enhancer was located ~139 kb upstream of the *SOX8*/*LMF1* promoter (Fig 4A). To

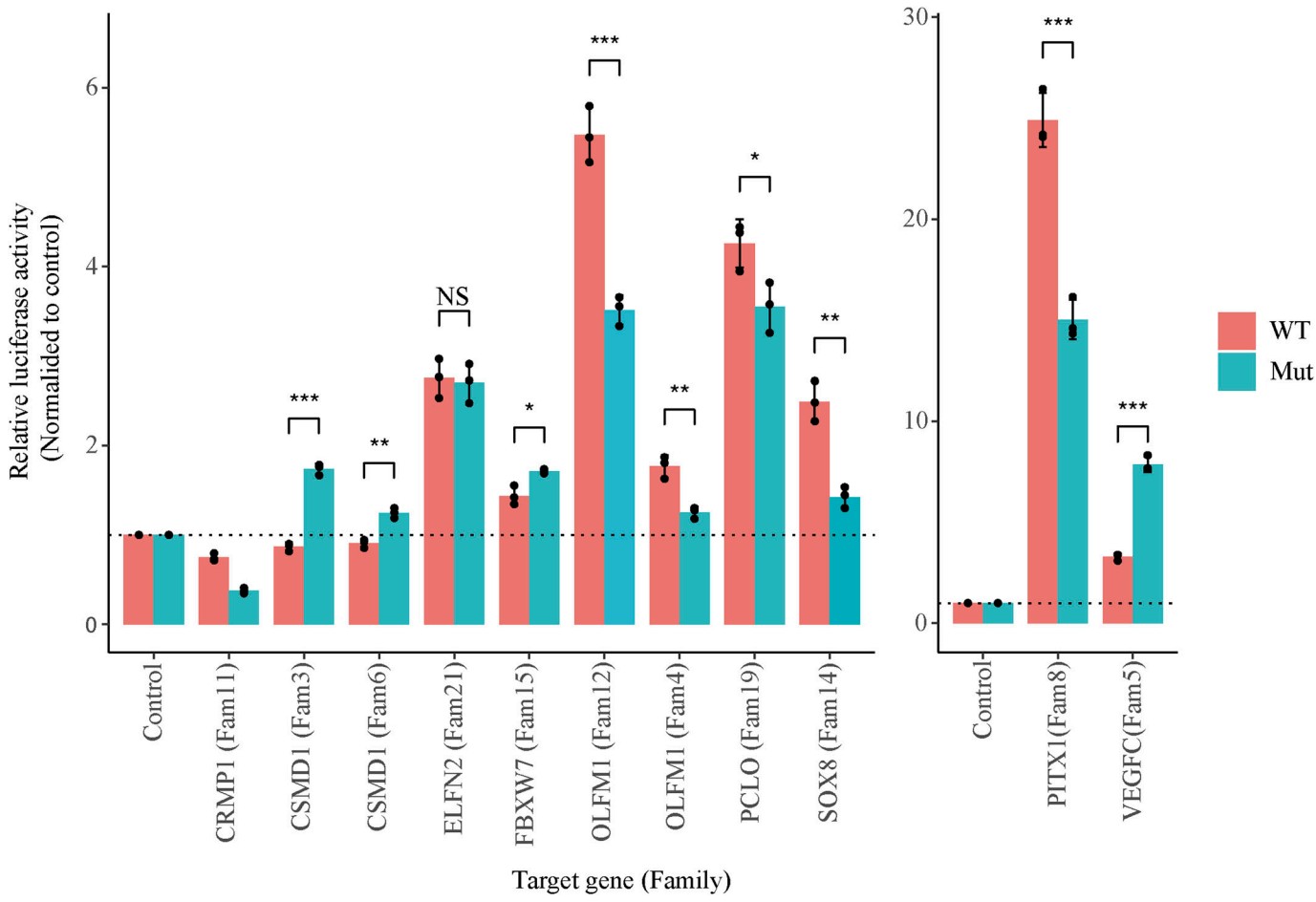

**Figure 3. Effect of DNM on enhancer activity.**
Dual-luciferase reporter assay of WT (reference) and the mutant (DNM) allele. The X-axis indicates the putative target genes of the enhancer, while the family IDs are shown in brackets. Y-axis indicates relative luciferase activity normalised to empty plasmid. The error bars indicate the SEM of three biological replicates. The enhancers associated with genes *PITX1* and *VEGFC* are plotted separately with different Y-axis scales because of the high activity of these enhancers. The significance level was calculated using a two-tailed *t* test. *** indicates *P*-value ≤ 0.001, ** indicate *P*-value between 0.01 and 0.001, whereas * indicates *P*-value between 0.01 to 0.05.

investigate whether the putative enhancer of the *SOX8*/*LMF1* gene indeed regulates the expression of the target genes, we performed CRISPR interference (CRISPRi), by guideRNA (Fig 4B) mediated recruitment of dCas9 fused with the four copies of sin3 interacting domain (SID4x) in the NPCs. We observed that CRISPRi of the *SOX8* enhancer led to the down-regulation of *SOX8* transcript levels in NPCs compared with non-target guideRNA controls (*P* = 0.034; Fig 4C). This suggests that the DNM-containing enhancer regulates the *SOX8* gene in neuronal cells.

We set out to perform the studies in neuroblastoma cells and validate the findings in NPCs. However, because of the difficulty in performing precise editing of a single nucleotide in neuroblastoma cells/NPCs, we have used HEK293T cells. The HEK293T cells show neuronal gene expression signature (41) and have been widely used for in vitro experiments to study neurodevelopment disorders (39). In addition, the DNM containing the *SOX8* enhancer was active in HEK293T cells (Fig 4F). To further investigate the direct impact of enhancer DNM on target gene expression in a genomic context, we knocked in enhancer DNM (chr18:g.893142:C>A) in the HEK293T cell

line using CRISPR/Cas9 (Fig 4D). In the heterozygous mutant clone, the *SOX8* gene showed a significant (*P* = 0.0301) reduction in expression levels; however, no difference was observed in the expression levels of the *LFM1* gene (*P* = 0.8641; Fig 4E), suggesting that the regulatory impact of DNM within the enhancer is specific for *SOX8* but not for *LFM1*.

The presence of mono-methylation of histone H3 at lysine 4 (H3K4me1) and acetylation of histone H3 at lysine 27 (H3K27ac) is a strong indicator of active enhancers. The histone mark H3K4me1 can be observed at both active and inactive enhancers but the H3K27ac mark is observed only at active enhancers. Hence, we investigated H3K4me1 and H3K27ac levels at DNM containing *SOX8* enhancer using ChIP–qPCR. We observed a significant reduction in H3K27ac levels at the *SOX8* enhancer region (*t* test *P* = 0.0099) in the mutant clone as compared with the WT. However, the level of H3K4me1 was not altered (*t* test *P* = 0.0674; Fig 4F). The significant reduction in H3K27ac, which is associated with enhancer activity at DNM containing the *SOX8* enhancer suggests a reduction in the enhancer activity upon

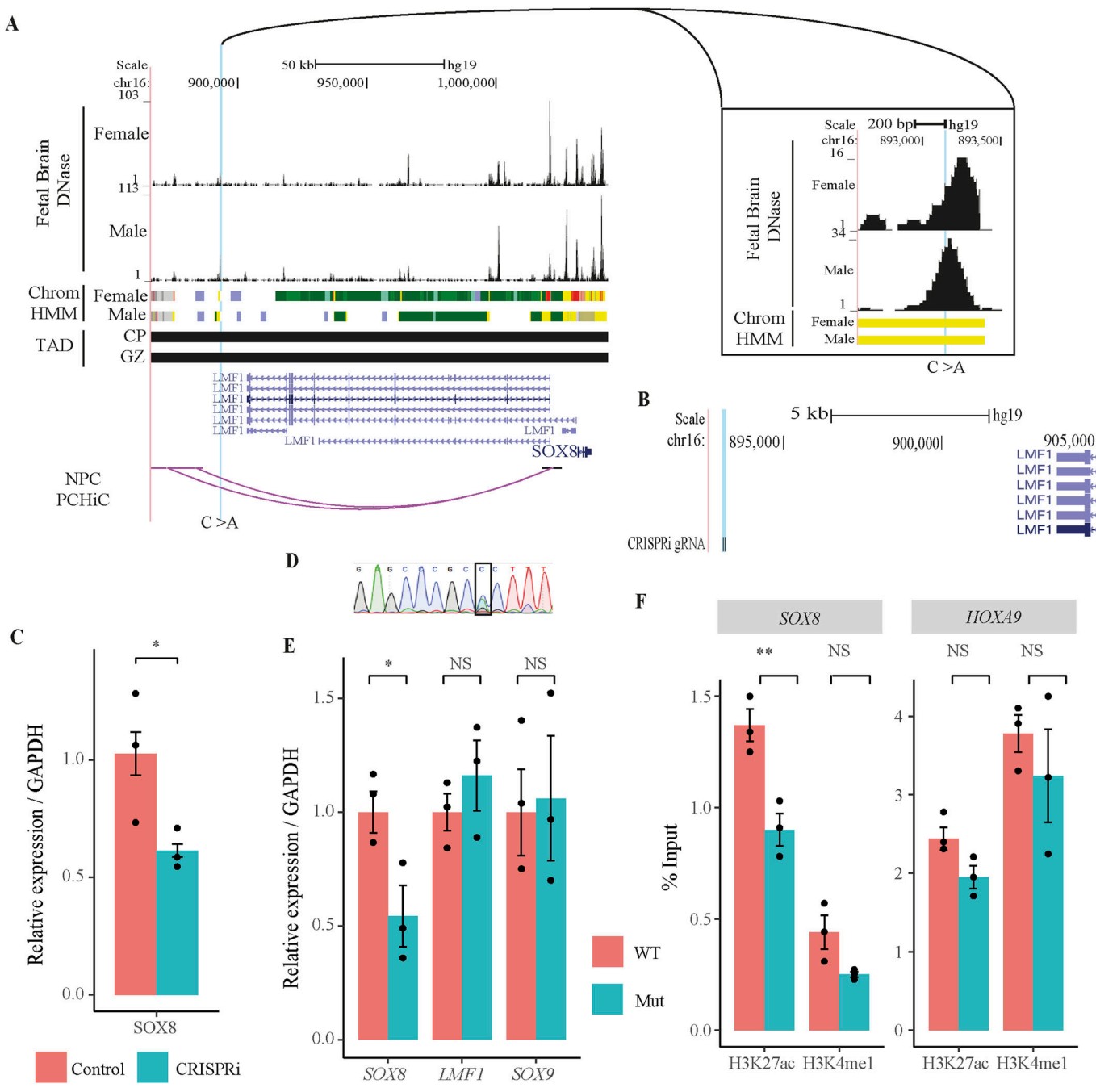

**Figure 4. Characterisation of *SOX8* enhancer DNM using CRISPR.**
**(A)** USCS tracks depicting male and female fetal brain DNase hypersensitivity peaks, ChromHMM tracks, fetal brain topologically associated domains, and enhancer–promoter interactions. Yellow bars in chromHMM tracks indicate enhancers. **(B)** USCS tracks depicting the location of CRISPRi gRNA with respect to the 3' end of the *LMF1* gene. **(C)** Relative expression levels of gene *SOX8* in neuronal progenitor cells with the CRISPRi of *SOX8* enhancer and non-target guide RNA controls, normalised to *GAPDH*. **(D)** Sanger sequencing trace file shows a heterozygous knock-in mutation in HEK293T cells using CRISPR/Cas9-mediated homology-directed repair. The black box highlights the location of the DNM. **(E)** Relative expression levels of *SOX8*, *LMF1*, and *SOX9* in WT and mutant cell line normalised to *GAPDH*. **(F)** Percentage of input (% input) of H3K27ac and H3K4me1 levels at *SOX8* enhancer and *HOXA9* promoter (control region) in WT and mutant cell lines. The significance level was calculated using a two-tailed *t* test. *** indicates *P*-value ≤ 0.001, ** indicate *P*-value between 0.01 and 0.001, whereas * indicates *P*-value between 0.01 to 0.05. WT, wild type; Mut, mutant.

DNM. At control loci (*HOXA9* promoter region), H3K27ac (*t* test *P* = 0.0751) and H3Kme1 (*t* test *P* = 0.444) levels were not altered in the mutant cells as compared with WT. We, therefore, conclude that

the enhancer harbouring DNM located 138,665 bp downstream of the *SOX8* promoter regulates the expression of *SOX8* in neuronal cells.

Taken together, our analysis suggests that the *SOX8* enhancer mutation from the family 14 ID proband indeed leads to reduced activity of the enhancers and, in turn, reduced the expression of the *SOX8* gene. Further experimental analysis is required to establish the *SOX8* enhancer DNM as a cause of ID in this patient. However, our study provides strong evidence that *SOX8* enhancer DNM might be a causal variant in this patient.

## Discussion

Despite the current widespread use of WGS, the true burden of pathogenic mutations in enhancers is unknown. This is mainly because of an inability to predict the pathogenicity of enhancer mutations based on sequence features. The aggregation of a minority of pathogenic mutations with the majority of benign regulatory mutations nullifies any signal from pathogenic mutations in non-protein-coding genomic regions in disease cohorts. It is noteworthy that in protein-coding regions of the genome, only protein-truncating variants and damaging missense variants, but not other protein-coding mutations, show significant enrichment in neurodevelopmental disorders (11, 42, 43). The analysis of DNMs in selected monogenic phenotypes provides a powerful analytic instrument because it can focus on a relatively small number of mutations that are more likely to be pathogenic. In this study, we show that DNMs in a cohort of patients with ID exhibit a non-random genomic distribution that differs from DNMs observed in healthy individuals, with several features consistent with a pathogenic role of noncoding DNMs. DNMs from patients with ID were thus selectively enriched in fetal brain enhancers, enhancers associated with genes that are ID-relevant, intolerant to loss of function mutations, and genes specifically expressed in the prefrontal cortex and disease-relevant transcription factor-binding sites.

Identifying genes that are recurrently mutated across multiple individuals has been a major route to discovering novel disease genes (44). We identified recurrent mutations within three fetal brain enhancer clusters associated with genes involved in nervous system development (*CSMD1, OLFM1, POU3F3*) and found that this enrichment was significant relative to expectations. One of them (*POU3F3*) was recently shown to harbour pathogenic heterozygous mutations in patients with ID (39). The remaining two genes *CSMD1* and *OLFM1* show high pLI scores (pLI = 1), indicating that they are intolerant to loss of function (LoF) mutations and are dosage-sensitive. More than 72% of the genes that are intolerant to loss of function mutations have not been assigned to any human diseases despite the strong evidence of constraint (27). It has been speculated that heterozygous loss of function mutations in these genes might be embryonically lethal; therefore, loss of function mutations in these genes might not be observed in the population (27). On the contrary, enhancers tend to be tissue, cell type, and developmental stage-specific; hence, the effects of enhancer mutations might manifest only in the tissues or the developmental stage at which the enhancer is active, leading to tissue-specific or developmental stage-specific disease phenotypes. We, therefore, hypothesise that DNMs in the tissue-specific enhancers of loss of function mutation intolerant genes might lead to disease even though the gene itself is not associated with any disease, thus unravelling novel gene–disease association.

Our study is underpowered to perform any statistical analysis because of the smaller cohort size. Hence, we explored a large ID cohort from Genomics England to find evidence in support of our findings. Indeed, we observed multiple ID probands with DNMs in *CSMD1, OLFM1,* and *POU3F3* enhancer clusters. Replicating our findings in a completely independent cohort provides strong support for our findings. In addition, we performed extensive experimental validation of the FBSE DNMs observed in our cohort. We performed a dual-luciferase reporter assay for one enhancer DNM each from our coding negative probands that contain FBSE DNM. We found that most enhancer DNMs tested show a significant effect on the enhancer activity. The effect of DNMs was in both directions with an almost equal number of DNMs showing gain and loss of enhancer activity.

Luciferase assay is an episomal assay; hence, we further investigated the effect of enhancer DNM on target gene expression in a genomic context. The *SOX8* is strongly expressed in embryonic and adult brains (45). We show that, upon CRISPRi of putative *SOX8* enhancer, transcript levels of *SOX8* were significantly reduced in NPCs suggesting that the DNM-containing enhancer regulates the *SOX8* gene in neuronal cells. In addition, we generated a heterozygous knock-in mutation at the putative *SOX8* enhancer in the HEK293T cell line. We show that the heterozygous variant at the putative *SOX8* enhancer region significantly reduces *SOX8* expression. This finding is particularly interesting as haploinsufficiency of *SOX8* has been implicated in ATR-16 syndrome characterised by α-thalassemia and intellectual disability (46, 47). The *SOX8* DNM may, therefore, be a potential cause of ID in a proband from our cohort.

Our work has integrated WGSs, epigenomics, and functional analysis to examine the role of regulatory DNMs in ID. Despite the genetic heterogeneity of ID, which severely hampers efforts to unequivocal demonstrate a causal role for individual non-coding mutations, our results provide multiple lines of evidence to indicate that functional regulatory mutations in stage-specific brain enhancers contribute to the aetiology of ID. This work should prompt extensive genetic analyses and mutation-specific experimental modelling to elucidate the precise role of regulatory mutations in neurodevelopmental disorders.

## Materials and Methods

### Selection criteria of intellectual disability patients for this study and ethical approval

The inclusion criteria for this study were that the affected individuals had a severe undiagnosed developmental or early-onset paediatric neurological disorder and that samples were available from both unaffected parents. Written consent was obtained from each patient family using a UK multicenter research ethics-approved research protocol (Scottish MREC 05/MRE00/74).

## Sequencing and quality control

WGS was performed on the Illumina X10 at Edinburgh Genomics. Genomic DNA (gDNA) samples were evaluated for quantity and quality using an AATI, Fragment Analyzer, and the DNF-487 Standard Sensitivity Genomic DNA Analysis Kit. Next-generation sequencing libraries were prepared using Illumina SeqLab-specific TruSeq Nano High Throughput library preparation kits in conjunction with the Hamilton MicroLab STAR and Clarity LIMS X Edition. The gDNA samples were normalised to the concentration and volume required for the Illumina TruSeq Nano library preparation kits, and then sheared to a 450-bp mean insert size using a Covaris LE220 focused-ultrasonicator. The inserts were ligated with blunt-ended, A-tailed, size selected TruSeq adapters and enriched using eight cycles of PCR amplification. The libraries were evaluated for mean peak size and quantity using the Caliper GX Touch with an HT DNA 1k/12K/HI SENS LabChip and HT DNA HI SENS Reagent Kit. The libraries were normalised to 5 nM using the GX data, and the actual concentration was established using a Roche LightCycler 480 and a Kapa Illumina Library Quantification kit and Standards. The libraries were normalised, denatured, and pooled in eights for clustering and sequencing using a Hamilton MicroLab STAR with Genologics Clarity LIMS X Edition. Libraries were clustered onto HiSeqX Flow cell v2.5 on cBot2s, and the clustered flow cell was transferred to a HiSeqX for sequencing using a HiSeqX 10 Reagent kit v2.5.

## Alignment and variant calling

De-multiplexing was performed using bcl2fastq (2.17.1.14), allowing 1 mismatch when assigning reads to barcodes. Adapters were trimmed during the de-multiplexing process. Raw reads were aligned to the human reference genome (build GRCh38) using the Burrows–Wheeler Aligner (BWA) mem (0.7.13) (48). The duplicated fragments were marked using samblaster (0.1.22) (49). The local indel realignment and base quality recalibration were performed using Genome Analysis Toolkit (GATK; 3.4-0-g7e26428) (50, 51, 52). For each genome, SNVs and indels were identified using GATK (3.4-0-g7e26428) HaplotypeCaller (53 Preprint), creating a gvcf file. The gvcf files of all the individuals from the same family were merged and re-genotyped using GATK GenotypeGVCFs producing a single VCF file per family.

## Variant filtering

Variant Quality Score Recalibration pipeline from GATK (50, 51, 52) was used to filter out sequencing and data processing artefacts (potentially false positive SNV calls) from true SNV and indel calls. The first step was to create a Gaussian mixture model by looking at the distribution of annotation values of each input variant call set that matches with the HapMap three sites and Omni 2.5 M SNP chip array polymorphic sites, using GATK VariantRecalibrator. Then, VariantRecalibrator applies this adaptive error model to known and novel variants discovered in the call set of interest to evaluate the probability that each call is real. Next, variants are filtered using GATK ApplyRecalibration such that the final variant call set contains all the variants with a 0.99 or higher probability to be a true variant call.

## DNM calling and filtering

The DNMs were called using the GATK Genotype Refinement workflow. First, genotype posteriors were calculated using sample pedigree information and the allele accounts from 1,000 genome sequence data. Next, the posterior probabilities were calculated at each variant site for each trio sample. Genotypes with genotype quality (GQ) < 20 based on the posteriors were filtered out. All the sites at which the parents' genotype and the child's genotype with GQs ≥ 20 for each trio sample were annotated as the high confidence DNMs. We identified, on average, 1,527 DNMs in 21 probands (~73 DNMs per proband) with GRCh38 assembly. Only high confident DNMs that were novel or had a minor allele frequency less than 0.001 in 1,000 genomes project were selected for further analysis. This resulted in the removal of 143 DNMs, resulting in a total of 1,384 DNMs (~66 DNMs per proband).

Because most of the publicly available datasets, including epigenomic datasets, are mapped to human genome assembly version hg19, we lifted over all the DNM coordinates to hg19 using the liftover package. The liftover resulted in the loss of 123 DNMs as they could not be mapped back to hg19 resulting in a set of 1,261 DNMs (60 DNMs per proband). All the variant coordinates presented in this article are from hg19 human genome assembly.

## DNM annotations

DNM annotations were performed using Annovar (54). To access DNM location with respect to genes, RefSeq, ENSEMBL, and USCS annotations were used. To determine allele frequencies, 1,000 genome, dbSNP, Exac, and GnomAD databases were used. To determine the pathogenicity of coding DNMs, annotations were performed with CADD, DANN, EIGAN, FATHMM, and GERP++ pathogenicity prediction scores. In addition, we determined whether any coding DNM has been reported in the ClinVar database as a pathogenic mutation.

## Structural variant detection and filtering

To detect structural variants (SV), we used four complementary SV callers: BreakDancer v1.3.6 (55), Manta v1.5.0 (56), CNVnator v0.3.3 (57), and CNVkit v0.9.6 (58). The BreakDancer and Manta use discordantly paired-end and split reads to detect deletions, insertions, inversions, and translocations, whereas CNVnator and CNVkit detect CNVs (deletions and duplications) based on read-depth information. The consensus SV calls were derived using MetaSV v0.4 (59). The MetaSV is the integrative SV caller which merges SV calls from multiple orthogonal SV callers to detect SV with high accuracy. We selected SVs that were called by at least two independent SV callers out of four.

To detect de novo SV, we used SV2 v1.4.1 (60). SV2 is a machine-learning algorithm for genotyping deletions and duplications from paired-end WGS data. In de novo mode, SV2 uses trio information to detect de novo SVs at high accuracy.

## Tissue-specific enhancer annotations

Roadmap Epigenomic Project (18) chromHMM segmentations across 127 tissues and cell types were used to define brain-specific

enhancers. We selected all genic (intronic) and intergenic enhancers (6_EnhG and 7_Enh) from a male (E081) and a female fetal brain (E082). This was accomplished using genome-wide chromHMM chromatin state classification in rolling 200-bp windows. All consecutive 200-bp windows assigned as an enhancer in the fetal brain were merged to obtain enhancer boundaries. A score was assigned to each enhancer based on the total number of 200-bp windows covered by each enhancer. Next, for each fetal brain enhancer, we counted the number of 200-bp segments assigned as an enhancer in the remaining 125 tissues and cell types. This provided enhancer scores across 127 tissues and cell types for all fetal brain enhancers. To identify FBSEs, Z scores were calculated for each fetal brain enhancer using the enhancer scores. Z scores were calculated independently for male and female fetal brain enhancers. Independent Z scores cutoffs were used for both male and female fetal brain enhancers such that ~35% of enhancers were selected. To define open accessible chromatin regions within brain-specific enhancers, we intersected enhancers with DNAse-seq data from the Roadmap Epigenomic Project (18) from a male (E081) and a female fetal brain (E082), respectively. Next, the male and female FBSEs were merged to get a final set of 27,420 FBSEs. We used a similar approach to identify enhancers that were specifically active in adult brain subsections, which include angular gyrus (E067), anterior caudate (E068), cingulate gyrus (E069), germinal matrix (E070), hippocampus middle (E071), inferior temporal lobe (E072), dorsolateral prefrontal cortex (E073), and substantia nigra (E074).

### Human gain enhancers

Human gain enhancers published previously by Reilly et al (13) were downloaded from Gene Expression Omnibus using accession number GSE63649.

### DNMs from healthy individuals

We downloaded DNMs identified in the healthy individuals in genomes of the Netherland (GoNL) study (20) from the GoNL website.

### Fetal brain-specific genes

Roadmap Epigenomic Project (18) gene expression (RNA-seq) data from 57 tissues was used to identify fetal brain-specific genes. We used female fetal brain gene expression data, as RNA-seq data were available only for the female fetal brain. For each gene, Z scores were calculated using RPKM values across 57 tissues. The genes with a Z score greater than two were considered brain specific.

### DNM enrichment analysis

The size of enhancer regions differs widely between tissues. Furthermore, the mutability of the tissue-specific enhancer region differs significantly. The mutability of FBSEs, human gain enhancers, and adult brain-specific enhancers was estimated using the previously defined framework for DNMs (34). The framework for the null mutation model is based on the tri-nucleotide context, where the second base is mutated. Using this framework, the probability of

mutation for each enhancer was estimated based on the DNA sequence of the enhancer. The probability of mutation of all the enhancers within the enhancer set (FBSEs, human gain enhancers, and adult brain subsections) was summed to estimate the probability of mutation for the entire enhancer set. The sequence composition and overall size in base pair vary significantly between fetal brain enhancers, human gain enhancers, and enhancers from adult brain subsections; hence, they may have different background mutation rates. To perform a valid comparison between the observed number of DNMs between fetal brain enhancers and adult brain enhancers, we normalised to the background mutation rate of fetal brain enhancers.

For example, the background mutation rate for FBSEs is 0.970718 and we observed 53 DNMs. Similarly, the background mutation rate for the adult brain subsection angular gyrus is 0.680226 and we observed 22 DNMs. Because of the difference in background mutation rate, we cannot directly compare the number of DNMs between fetal brain enhancers and angular gyrus. Hence, we normalised the observed number of DNMs in angular gyrus enhancers to a background mutation rate of 0.970718 using the following formula.

$$\frac{\text{(Observed number of DNMs in angular gyrus enhancers} \times \text{mutation rate of fetal brain enhancers)}}{\text{mutation rate of angular gyrus enhancers,}}$$

$$\frac{(22 \times 0.970718)}{0.680226} = 31.395$$

Similarly, we normalised the observed number of mutations from all adult brain subsections and human gain enhancers to a background mutation rate of 0.970718 (Table S6) so that we could perform a valid comparison between the observed number of mutations from various enhancer sets. The significance level between DNMs observed in FBSEs (n = 2) and adult brain enhancers (n = 8) was calculated using a two-tailed $t$ test.

### Enrichment of recurrently mutated enhancer clusters

The enhancer clusters were randomly shuffled 1,000 times. We estimated the number of enhancer clusters with more than one mutation for each iteration. Then, we counted the number of times when more than or equal to two mutations were observed in three or more enhancer clusters. This number was then divided by 1,000 to calculate the $P$-value.

### DNM effect on transcription factor binding

The R Bioconductor package motifbreakR (61) was used to estimate the effect of DNM on transcription factor-binding. The motifbreakR works with position probability matrices for transcription factors (TF). MotifbreakR was run using three different TF databases: viz. homer, encodemotif, and hocomoco. To avoid false TF-binding site predictions, either with the reference allele or with an alternate allele, a stringent threshold of 0.95 was used for motif prediction. DNMs that create or disturb a strong base (position weight ≥ 0.95) of

the TF motif, as predicted by motifbreakR, were selected for further analysis.

### Prediction of target genes of enhancers

Three different methods were used to predict the potential target genes of enhancers.

Chromosome conformation capture (Hi-C) comprehensively detects chromatin interactions in the nucleus; however, it is challenging to identify individual promoter–enhancer interactions using Hi-C because of the complexity of the data. In contrast, promoter capture Hi-C (PCHi-C) specifically identifies promoter–enhancer interactions as it uses sequence capture to enrich the interactions involving promoters of annotated genes (62). The significant interactions between promoters and enhancers identified using PCHi-C in neuronal progenitor cells (21) were used to assign target genes to the DNM-containing enhancers. The enhancers overlapped with the PCHi-C HindIII fragments. If an overlap was found between an enhancer and the PCHi-C HindIII fragment, the significantly interacting regions (PCHi-C HindIII fragments representing promoters of the genes) of the PCHi-C HindIII fragment were extracted to assign genes to the enhancers.

For an enhancer to interact with a promoter, both promoter and enhancer need to be active in specific cells at a specific stage. To identify promoter–enhancer interactions, all the active promoters in the fetal brain (as defined by chromHMM segmentation) were extracted. Promoter–enhancer interactions occur within TAD; hence, promoters located within the same TAD as that of a DNM-containing enhancer were used for analysis.

For each enhancer and promoter, H3K27ac counts were extracted from all tissues for which H3K27ac data were available in the Roadmap Epigenomic Project (18) ChIP-seq dataset. For the fetal brain, H3K27ac ChIP-seq data published by Reilly et al (13) were used because H3K27ac ChIP-seq data were not available in the Roadmap Epigenomics Project ChIP-seq dataset for the fetal brain. The Spearman rank correlation coefficient (Spearman's rho) was calculated between each enhancer–promoter pair within the TAD using Scipy stats.spearmanr function from Python. The permutation test was performed to identify the significance of the correlation. The counts were randomly shuffled, independently for enhancers and promoters, 1,000 times to calculate an adjusted *P*-value. The interactions with an adjusted *P*-value less than 0.01 were considered a significant interaction between the enhancer and promoter.

Finally, if any enhancers remained unassigned to a gene using these approaches, they were assigned to the closest fetal brain-expressed genes within the TAD. A gene with an expression level more than or equal to 1 TPM in the Roadmap Epigenomics Project fetal brain RNA-seq data was considered to be expressed in the fetal brain.

### Enrichment analysis for known ID genes

To test if enhancer-associated genes were enriched for genes previously implicated in neurodevelopmental disorders, three different gene sets were used: (1) intellectual disability (ID) gene list published in the review by Vissers et al (1) was downloaded from the Nature website; (2) we compiled all the genes implicated in neurodevelopmental disorders in the DDD project (4); and (3) all the genes implicated in ASD were downloaded from the SFARI browser. The significance of enrichment was tested using a hypergeometric test in R.

### Gene ontology enrichment analysis and tissue enrichment analysis for genes associated with DNM-containing enhancers

Gene ontology enrichment and tissue enrichment analysis for genes associated with the DNM-containing enhancer were performed using the web-based tool Enrichr (http://amp.pharm.mssm.edu/Enrichr/). The list of genes was uploaded to the Enricher web interface. Enrichr enables users to submit lists of human or mouse genes to compare against numerous gene set libraries of known biological function, such as pathways, diseases, or gene sets regulated by transcription factors. The matching gene sets are ranked by different methods that assess the similarity of the input gene set with all other gene sets in each library (63).

### Enrichment of analysis for tissue/cell-type expression of transcription factors whose binding sites were affected by enhancer DNM

The analysis was performed using the web-based tool Enricher (http://amp.pharm.mssm.edu/Enrichr/). The list of genes was uploaded to the Enricher web interface. Enricher uses the data from the ARCHS4 project (64) to estimate the enrichment of genes that are expressed in specific tissues and cell types. ARCHS4 contains processed RNA-seq data from over 100,000 publicly available samples profiled by two major deep sequencing platforms HiSeq 2000 and HiSeq 2500.

### Enrichment analysis for pLI scores

The probability of loss of function intolerance (pLI) scores for each gene was downloaded from Exome Aggregation Consortium (ExAC) browser (https://gnomad.broadinstitute.org/). The significance of enrichment was tested using a hypergeometric test in R.

### Regulatory regions of developing human cortex and developed human prefrontal cortex cell types

The developing human cortex cell type ATAC-seq and PLAC-seq data published in (30) were downloaded from the Neuroscience Multi-Omic Archive (NeMO Archive). The human prefrontal cortex cell type enhancer and promoter annotations were downloaded from (28). DNMs were overlapped with regulatory regions using bedtools intersectBed (65). Significance of enrichment was calculated using Fisher's exact test in R.

### Cell culture

Neuroblastoma cell line (SH-SY5Y) was maintained in DMEM/F12 media (Gibco), 1% penicillin–streptomycin, 10% fetal bovine serum,

and 2 mM L-glutamine. The HEK293T cells were maintained in DMEM, 10% FCS, and 1% penicillin–streptomycin.

## Dual-luciferase enhancer assays

Enhancer and control regions (500–600 bp) were amplified from human genomic DNA from HEK293T cells using Q5 High-Fidelity Polymerase (New England Biolabs). Amplified fragments were cloned into pGL4.23 plasmid (Promega), which consists of a minimal promoter and the firefly luciferase reporter gene. These regions were mutagenised to introduce the DNMs of interest using the Q5 Site-Directed Mutagenesis kit (New England Biolabs) using non-overlapping primers. pGL4.23 plasmids containing putative enhancer DNA were sequence-verified and transfected, together with a Renilla luciferase–expressing vector (pRL-TK; Promega) into SHSY-5Y cells using Lipofectamine 3000 (Invitrogen) following the manufacturer's protocol. Firefly and Renilla luciferase activities were measured 24 h after transfection using the Dual-Luciferase Reporter Assay System (Cat. number E1910; Promega) as per the manufacturer's instructions. Primers used to amplify genomic DNA and for mutagenesis are provided in Table S14.

## Genome editing in HEK293T cells

To generate HEK293T cells carrying the mutation at the putative *SOX8* enhancer element, cells were co-transfected with gRNA (Fig S1) expression plasmid (1 $\mu$g/ml) and the repair template with PAM mutation only (control) or repair template (1 $\mu$l of 10 $\mu$M) with both PAM mutation and enhancer variants using Lipofectamine 3000 transfection reagent (Thermo Fisher Scientific) as per the manufacturer's instructions. After 48 h, successfully transfected cells were selected by puromycin treatment (2.5 $\mu$g/ml) for 48 h. The resulting puromycin-resistant cells were plated at 5,000 cells/ 10 cm$^2$. After 1 wk, colonies were picked and plated in duplicate at 1 colony/well in a 96-well plate. Genomic DNA was extracted from the colonies and sequenced by Sanger sequencing. WT clones with PAM mutation only and heterozygous *SOX8* enhancer variant were expanded and frozen for later use. The *SOX8* enhancer sequencing primers are provided in Table S15.

## Genomic DNA extraction

Genomic DNA was extracted by a modified version of the salting-out method. Briefly, cells were lysed in a lysis buffer (100 mM Tris–HCl pH 8.5; 5 mM EDTA; 200 mM NaCl; 0.2% SDS) plus 4 U/ml of Proteinase K (Thermo Fisher Scientific) for at least 2 h at 55°C with agitation. Then, 0.4× vol of 5 M NaCl were added to the mixture and centrifuged at max. speed for 10 min. DNA in the supernatant was precipitated with 1× vol of isopropanol. After centrifugation, the pellet was washed with 70% ethanol and air-dried for half an hour. The DNA was resuspended in water and incubated for at least 1 h at 37°C with agitation.

## RNA isolation, cDNA synthesis, and RT–qPCR

RNA was extracted using the RNeasy Mini Kit (QIAGEN), and cDNA was produced with the First Strand cDNA Synthesis Kit (LunaScript RT SuperMix Kit; New England Biolabs). qPCR reactions were performed with SYBR Green Master Mix (Luna Universal qPCR Master Mix; New England Biolabs) and run on a CFX96 Real-Time PCR machine (Bio-Rad). Relative gene expression values were calculated with the −ΔCt method, using GAPDH as a housekeeping gene for normalisation. Oligonucleotides used for qPCR are provided in Table S16.

## NPC culture protocol

H9 human embryonic stem cells hESCs (WiCELL) were differentiated to NPCs by adding Gibco PSC Neural Induction Medium (A1647801; Thermo Fisher Scientific). After 7 d of induction, the cells were passaged in an NPC proliferation medium containing Advanced DMEM/F-12 - 24.5 ml (12634010; Thermo Fisher Scientific), Neurobasal – 24.5 ml (GIBCO Neural Induction Supplement).

## CRISPR interference (CRISPRi) with dCAS9-Sid4x

CRISPRi using dCAS9-Sid4x is performed as previously described in (66) with the following modifications. Oligos with guide RNA sequences (Table S17) were cloned into Addgene plasmid ID pSLQ1371 (kind gift by Stanley Qi) following the protocol previously described by (67). pSEQ1371 plasmid was used as a non-targeting control.

For NPCs transfection, 7.5 × 10$^5$ NPCs were plated onto one well of a six-well plate precoated with geltrex (A1413302; Gibco) in the NPC proliferation medium (49% neurobasal medium [A1647801; Gibco], 49% advanced DMEM/F12 [12634010; Gibco], and 2% of neural supplement [A1647801; Gibco]). ~24 h after plating, 625 ng of respective guide RNAs were diluted into 250 $\mu$l opti-MEM (31985062; Gibco) and 1875 ng dCAS9-pSid4X. 10 $\mu$l of TransIT-X2 reagent (MIR6004; Mirus Bio) was added to the above mix and thoroughly mixed and incubated at room temperature for 20 min. 260 $\mu$l of the final transfection mix was then added onto one well of a six-well plate containing 2.5 ml media. The plate was rocked to mix and incubated for 24 h. 48 h after transfection, 0.5 $\mu$g/ml puromycin (A1113803; Gibco) was added to the media. Cells were harvested with accutase (A1110501; Gibco) after 24 h of puromycin selection and taken for RNA isolation and RT-qPCR.

For CRISPRi in NPC-derived neuronal cells, 10$^5$ NPCs were plated onto one well of a 24-well plate precoated with 100 $\mu$g/ml PLO (P4957; Sigma-Aldrich) and 5 $\mu$g/ml lamin (L2020; Sigma-Aldrich). ~24 h after plating, NPC proliferation medium was replaced with 750-500 $\mu$l neuronal maturation medium (BrainPhys Neuronal Medium [05790; Stemcell], NeuroCult SM1 Neuronal Supplement [05711; Stemcell], N2 Supplement-A [07152; Stemcell], 20 ng/ml Human Recombinant BDNF 78005; Stemcell], 20 ng/ml Human Recombinant GDNF [78058; Stemcell], 1 mM Dibutyryl-cAMP [D0627; Sigma-Aldrich], and 200 nM Ascorbic Acid [072132; Stemcell]). 3 d after the start of neuronal differentiation, 200 ng of the respective guide RNAs were diluted into 50 $\mu$l opti-MEM and 600 ng of dCAS9-pSid4X. 2 $\mu$l of TransIT-X2 reagent was added to the above mix and thoroughly mixed and incubated at room temperature for 20 min. 52 $\mu$l of the final transfection mix was then added onto one well of a 24-well plate containing 0.5 ml neuronal maturation medium. The plate was rocked to mix everything and incubated for 24 h. 24 h after transfection, 0.5 $\mu$g/ml puromycin was added to the

media. Surviving transfected cells after puromycin selection were harvested with accutase after 48 h of antibiotic selection and taken for RNA isolation and qPCR.

## Statistical analysis

All luciferase experiments, gene quantification using qPCR, and H3K4me1 and H3K27a levels using ChIP–qPCR were done in three biological replicates. The significance level was calculated using a two-tailed $t$ test.

## False discovery rate calculation

We used R package p.adjust to calculate false discovery rate (FDR) to correct for the multiple hypothesis testing.

# Data Availability

The WGS data are not publicly available because they contain information that could compromise research participant privacy/consent. However, WGS data and variant calls that support the findings of this study are available on request from the corresponding author (SS Atanur).

# Supplementary Information

# Acknowledgements

We thank the families of the affected children for their time and support for the research. We thank Prof Andrew Jackson for helpful discussions and for obtaining ethical approval for the study. We thank Mrs Sophie Shi for contributing to reagent generation. We also thank Dr Patrick Short and Dr Kaitlin Samocha, both from Sanger Institute, for providing a trinucleotide probability table and helpful discussion on a mutational model, respectively. This research was made possible through access to the data and findings generated by the 100,000 Genomes Project. The 100,000 Genomes Project is managed by Genomics England Limited (a wholly owned company of the Department of Health and Social Care). The 100,000 Genomes Project is funded by the National Institute for Health Research and NHS England. The Wellcome Trust, Cancer Research UK, and the Medical Research Council have also funded research infrastructure. The 100,000 Genomes Project uses data provided by patients and collected by the National Health Service as part of their care and support. This research was supported by the National Institute for Health Research (NIHR) Imperial Biomedical Research Centre. This work was funded by grants from the Wellcome Trust Institute Strategic Support and National Institute for Health Research (NIHR) Imperial Biomedical Research Centre, Institute for Translational Medicine and Therapeutics (P70888) obtained by SS Atanur. J Ferrer and MG De Vas's work was funded by grants from the Wellcome Trust (WT101033 to J Ferrer), Medical Research Council (MR/L02036X/1 to J Ferrer), and European Research Council Advanced Grant (789055 to J Ferrer). MM Pradeepa's lab is funded by the UKRI/MRC (MR/T000783/1), and Barts charity (MGU0475) grants. TN Khan was partially supported by the Government of Pakistan under the PSDP project "Development of National University of Medical Sciences (NUMS), Rawalpindi."

## Author Contributions

MG De Vas: validation, investigation, and writing–review and editing.
F Boulet: validation, investigation, and writing–review and editing.
SS Joshi: data curation and formal analysis.
MG Garstang: validation.
TN Khan: validation.
G Atla: formal analysis.
D Parry: formal analysis.
D Moore: resources.
I Cebola: validation.
S Zhang: validation.
W Cui: validation.
AK Lampe: resources.
WW Lam: resources.
J Ferrer: validation and writing–review and editing.
MM Pradeepa: supervision, validation, and writing–review and editing.
SS Atanur: conceptualization, data curation, software, formal analysis, supervision, funding acquisition, investigation, visualization, methodology, project administration, and writing–original draft, review, and editing.

## Conflict of Interest Statement

The authors declare that they have no conflict of interest.

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
