## [Reviewer comments · Life Science Alliance]

Life Science Alliance

Regulatory de novo mutations underlying intellectual disability

Santosh Atanur, Matías de Vas, Fanny Boulet, Shweta Joshi, Myles Garstang, Tahir Khan, Goutham Atla, David Parry, David Moore, Ines Cebola, Shuchen Zhang, Wei Cui, Anne Lampe, Wayne Lam, Jorge Ferrer, and Pradeepa Madapura

DOI: <https://doi.org/10.26508/lsa.202201843>

Corresponding author(s): Santosh Atanur, Imperial College London and Pradeepa Madapura, Queen Mary London University / Blizard Institute

Review Timeline:

Submission Date:	2022-11-21
Editorial Decision:	2023-01-13
Revision Received:	2023-02-04
Editorial Decision:	2023-02-08
Revision Received:	2023-02-11
Accepted:	2023-02-13

Transaction Report:

Please note that the manuscript was reviewed at Review Commons and these reports were taken into account in the decision-making process at Life Science Alliance.

Manuscript number: RC-2022-01656

Corresponding author(s): Santosh, Atanur

1. General Statements [optional]

We thank all the reviewers for the constructive comments to improve the overall quality of the manuscript. We carefully reviewed all the reviewers' comments and addressed them to the best of our ability. We provided additional methodological details of the enrichment analysis and provided additional clarity about the computational and experimental analysis presented in the manuscript. We have addressed all the reviewers' concerns. Below we provide our point-to-point responses and hope that they are satisfactory.

Reviewer #1 (Evidence, reproducibility and clarity (Required)):

Comment 1.1: Figure 4. The figure legend and sub-figures are inconsistent. They do not match.

Response 1.1: Apologies for the error. In the revised manuscript we have changed the order of panels in Figure 4 to make it consistent with the figure legends.

Comment 1.2: Figure 4. For the Sanger sequencing trace of the edited HEK293 cells, why there are noise peak?

Response 1.2: It is a heterozygous knock-in, with only one allele has a mutation. Moreover, it is a PCR product we have sequenced hence it looks noisy.

Comment 1.3: How many single cell clones were chosen for further analyses after CRISPR genome editing? The authors should do single cell filtering by Flow Cytometer or others.

Response 1.3: We had one clone with heterozygous knock-in.

Comment 1.4: The authors conducted RT-qPCR to quantify mRNA expression, RNA-Sequencing should be more accurate.

Response 1.4: We had one clone with heterozygous knock-in hence we used this clone for RT-qPCR. As reviewer no 3 suggested, RNA sequencing is not needed to show the effect of this mutation on genes in cis.

Comment 1.5: The discussion is too long, please shorten.

Response 1.5: In the revised manuscript we have shortened the discussion.

CROSS-CONSULTATION COMMENTS

I agree with the other reviewers' comments.

Reviewer #1 (Significance (Required)):

This study investigates the genetic and molecular mechanisms of intellectual disability (ID) by integrating whole genome sequencing and follow up functional explorations. The results provide novel insights into genetic aetiology of ID.

Reviewer #2 (Evidence, reproducibility and clarity (Required)):

The manuscript by De Vas et al describes an investigation of the contribution of non-coding de novo variants to intellectual disability (ID). The authors perform whole genome sequencing (WGS) of 21 ID probands and both parents, and combine these data with WGS from 30 trios previously sequenced. The authors use publicly available data from the Roadmap Epigenomics project to identify sets of enhancers hypothesised to have a role in ID, such fetal brain specific enhancers and enhancers associated with known ID-associated genes. These enhancer sets are then tested for enrichment of non-coding de novo variants ID, using publicly available de novo variant data from the Genome of Netherlands (GoNL) project as a control comparison. The authors report that de novo variants in ID are significantly enriched within fetal brain-specific and human-gained enhancers. This is perhaps the main finding of the study. The authors also identify recurrent de novo variants in ID within clusters of enhancers that regulate the genes CSMD1, OLDM1 and POU3F3 in ID. A number of functional experiments are performed to provide further insights in the mechanisms by which de novo variants impact the expression of putative target genes; for example, data is provide that show de novo variants observed in ID within a SOX8 enhancer leads to reduced expression of the SOX8 gene. In conclusion, the authors claim that their data support de novo variants in fetal brain enhancers as contributing to the aetiology of ID.

Major comments.

The study uses leading edge genomic technologies to generate WGS in a new ID sample, which is used to investigate the role of non-coding variants to ID aetiology. The manuscript is in general very well written. However, a weakness of the study is a very small sample size, which should result in low statistical power. Despite this power consideration, the authors report very strong P values for their main findings. My main concern with the study is that the methodology used to evaluate enrichment of de novo variants within specific sets of enhancers is unclear, and therefore as it currently stands, I am unable to be confident in the findings. I am also concerned about whether data from the Genome of the Netherlands project is a suitable control comparison, given technical differences that are likely to exist between this and the ID data set. I further explain these methodological concerns below:

Comment 2.1: When testing for the enrichment of de novo variants, the most commonly used approach in the field involves testing whether the observed number of de novo variants in a given genomic region is greater than the number expected by chance, using a Poisson test. Here,

the expected number of de novo variants is derived from trinucleotide mutation rates. This method was first proposed by Samocha et al 2014. The current authors use trinucleotide mutation rates to estimate the expected number of de novos among enhancer sets, and cite the Samocha paper, but my understanding is that they do not use a Poisson test to evaluate enrichment. Instead, they use the expected number of mutations among the enhancer sets to normalise the observed number of de novo variants, but it is not clear to me why this is performed, and also what data and the statistical test is actually being used to evaluate de novo variant enrichment? I can guess at what they have done, but the methods section outlining this test should be more clearly explained.

Response 2.1: The Samocha et al 2004 paper provides a statistical framework to estimate the expected number of DNMs under neutral evolution. However, our aim was not to estimate the enrichment of DNM in fetal brain enhancers with a background rate of mutation (see the answer to the next comments for a detailed explanation). Our aim was to investigate whether in our ID cohort DNMs were enriched in the enhancers that are specifically active in the fetal brain or the enhancers that are active in specific subsections of the adult brain. Hence, we compared the number of DNMs in the fetal brain enhancers (Fetal brain enhancers and human gain enhancers) with the number of DNMs in enhancers of various sub-sections of the adult brain. In Table S6 of the revised manuscript, we have highlighted the values that were used for the statistical test. We used a T-test to estimate whether fetal brain enhancers were enriched for ID DNM as compared to adult brain enhancers.

However, as pointed out by the reviewer in comment 2.3, the sequence composition and overall size in base pair vary significantly between fetal brain enhancers, human gain enhancers and enhancers from adult brain subsections thus they may have different background mutation rates. Hence, before doing any comparison between DNMs in various enhancer sets (fetal vs adult), it is important to normalise them to the same background mutation rate for valid comparison. Hence, we used the framework provided in Samocha et al 2014 paper to estimate the background mutation rate of various enhancer sets and normalised them to the background mutation rate of fetal brain-specific enhancer set.

For example, the background mutation rate for fetal brain-specific enhancers is 0.970718 and we observed 53 DNMs. Similarly, the background mutation rate for the adult brain sub-section angular gyrus is 0.680226 and we observed 22 DNMs. Because of the difference in background mutation rate, we cannot directly compare the number of DNMs between fetal brain enhancers and angular gyrus. Hence, we normalised the observed number of DNMs in angular gyrus enhancers to a background mutation rate of 0.970718 using the following formula.

(Observed number of DNMs in angular gyrus enhancers x mutation rate of fetal brain enhancers) / mutation rate of angular gyrus enhancers

$$(22 \times 0.970718) / 0.680226 = 31.395$$

Similarly, we normalised the observed number of mutations from all adult brain subsections and human gain enhancers to a background mutation rate of 0.970718 (Table S6) so that we could

Full Revision

perform a valid comparison between the observed number of mutations from various enhancer sets.

In the revised manuscript, we have revised the method section to make it clearer (Page 23, line 23 to page 24, line 19).

Comment 2.2: Can the authors please explain why they did not use the standard *de novo* variant enrichment approach outlined in Samocha et al 2014, which is used in similar non-coding *de novo* studies of ID (e.g. Short et al 2018 Nature)? My concern is that using the Samocha approach, no enrichment would be observed in fetal brain enhancers, given the data presented in supplemental table S6.

Response 2.2: The Samocha et al 2014 paper provides the statistical framework to evaluate the rates of *de novo* mutation (DNM) assuming neutral selection. The variants that lead to disease (functional variants) tend to be under negative selection. Thus, the region or a gene that is devoid of functional variants is likely to reflect a region or a gene that is under selective constraint. The functional variants in such regions or genes are likely to cause disease (Samocha et al, 2014, Nature Genetics). This approach was used to identify genes that are intolerant to loss of function mutations (Lek et al, 2016, Nature).

As we discussed in the manuscript, due to the triplet codon structure it is relatively easy to predict functional consequences of DNMs in protein-coding regions of the genome, thus it becomes easy to distinguish likely functional variants from non-functional variants. Please note that only protein-truncating and damaging missense (potentially functional) coding DNMs show enrichment in NDD and not non-functional DNMs.

In non-coding regions of the genome, in absence of a codon like structure, it is extremely challenging to distinguish potentially functional variants from non-functional variants. A very small proportion of the DNMs that overlap enhancer regions might be truly functional (under selective pressure) and the majority might be non-functional (neutral). Hence, it is not possible to achieve statistical significance using Samocha et al 2014 framework for enhancer DNMs with a small cohort when the enhancer set contains a mixture of functional (a small fraction) and non-functional (a large fraction) DNMs. An analogy for the protein coding region would be applying Samoch et al 2014 framework to all protein-coding variants including synonymous mutations, which may not show enrichment of DNMs in the disease cohort.

Given the small sample size and non-availability of tools and techniques to separate functional non-coding variants from non-functional variants, we did not use Samocha et al 2014 framework to show the enrichment of DNMs in fetal brain enhancers. Instead, we asked a simple question, out of fetal and adult brain enhancer sets which one is enriched for DNMs in the ID cohort?

In the revised manuscript, in the abstract (Page 3, line 8) we changed the sentence to clarify that the enrichment of ID DNMs in fetal brain enhancers was against the adult brain enhancers.

Comment 2.3: In Supplemental table S6, the normalised expected number of *de novo* variants across all different enhancer sets within the ID and GoNL samples is the same. Can the authors

Full Revision

clarify why this is the case, as presumably these sets contain very different genomic sequences, and therefore one would not expect the same number of DNMs?

Response 2.3: See the detailed explanation in answer to comment 2.1. As we normalised observed the number of DNMs from various enhancer sets to the background mutation rate of fetal brain enhancers (0.970718), the expected number of DNMs (number of samples X mutation rate, $47 \times 0.970718 = 45.623746$) is the same for all enhancer sets.

Comment 2.4: Instead of using the standard enrichment approach proposed by Samocha et al 2014, the authors compare the rates of de novo variants in ID to those reported in the GoNL study. However, very little information is provided about the de novo variant data from the GoNL. Presumably, the GoNL and the current study used different approaches to sequence samples, call variants, and QC the data. Also, is the coverage across these studies comparable? All these factors will contribute to batch effects, and therefore I am not convinced that the GoNL study is an appropriate control comparison. The authors should provide data to reassure the reader that these samples can be compared. For example, are similar rates of de novo variants found between these samples for variants in null enhancers sets? To clarify, an equivalent analysis in exome sequencing studies would be to show that the rates of synonymous variants are the same across data sets.

Response 2.4: We would like to point out that we haven't performed a direct comparison between our ID cohort and GoNL cohort. We are aware that there are technical differences between DNM identification in our cohort and the GoNL cohort. The GoNL genomes were sequenced on Illumina HiSeq 2000 with 13X coverage while ID cohort reported in this study were sequenced on the Illumina HiSeq X10 platform with an average coverage of 37X. **Hence, We did not perform a direct comparison between our ID cohort and GoNL cohort.**

We evaluated the enrichment of DNMs in fetal brain-specific enhancers compared to adult brain-specific enhancers independently within ID and GoNL cohorts. We compared the number of DNMs in fetal brain enhancers vs adult brain enhancers within the ID cohort. We observed the significant enrichment of DNMs in fetal brain-specific enhancers as compared to adult brain enhancers in the ID cohort. Next, we asked whether the DNMs from healthy individuals also show enrichment in fetal brain-specific enhancers or whether this enrichment was specific to the ID cohort. To answer this question, we used the GoNL cohort and performed a comparison between fetal brain enhancers and adult brain enhancers within GoNL cohort. We did not find any enrichment in fetal brain enhancers. As analysis is performed independently within each cohort between fetal and adult brain enhancers, hence the technical differences between the two datasets would not have any effect on the results.

To make it clear, we have changed the text in the revised manuscript (Page 8, lines 1-4). We have also changed a sentence in the abstract from “We found that regulatory DNMs were selectively enriched in fetal brain-specific and human-gained enhancers.” to “We found that regulatory DNMs were selectively enriched in fetal brain-specific and human-gained enhancers as compared to adult brain enhancers.”

Comment 2.5: The replication analysis of enhancer clusters that are recurrently hit by de novo

Full Revision

variants in ID is weak. For enhancer clusters with recurrent de novo variants in their ID cohort, the authors simply report the number of de novo variants observed in these enhancers in the Genomics England cohort, but they do not test whether the observed number in Genomics England is greater than that expected. For their findings to be replicated, they need to show the de novo rate is statistically above expectation.

Response 2.5: To improve the replication analysis, we estimated the expected number of DNMs in the Genomics England cohort (n=3,169) in *CSMD1*, *OLFM1* and *POU3F3* enhancer clusters using the framework defined in Samocha et al 2014 paper and estimated statistical significance using a poisson test. We found that the *POU3F3* enhancer cluster was significantly enriched for DNMs even after multiple test corrections. We included these findings in the revised manuscript (Page 12, lines 24-27). In addition, we applied Samocha et al framework to *CSMD1*, *OLFM1* and *POU3F3* enhancer clusters in our ID cohort as well. We found that all three enhancer clusters were enriched for DNMs after multiple test correction.

Minor comments:

Comment 2.6.1: The authors state that all coding de novos were validated by Sanger sequencing, but what about the non-coding de novos? Validation of the specific mutations that contribute to the main findings would strengthen the paper.

Response 2.6.1: The potentially pathogenic coding variants were validated using sanger sequencing by clinicians to report our findings to respective families. However, as non-coding DNMs could not be reported back to families as a diagnosis until the pathogenicity of these DNMs is fully established, clinicians (who have patients' DNA) are reluctant to perform Sanger sequencing to confirm the DNM. However, we have investigated each non-coding variant reported in the manuscript in IGV and their pattern looks similar to the validated coding DNMs, hence we are confident that they are true DNM calls.

Comment 2.6.2: In the introduction, the line "A family with two affected siblings was analysed for the presence of recessive variants" seems out of place and incomplete, as there is no mention of the results from this analysis.

Response 2.6.2: Sorry for the error, we have removed this sentence from the manuscript.

Comment 2.6.3: In the discussion, they write "It is noteworthy that in protein-coding regions of the genome, only protein-truncating variants (PTV), but not other protein-coding mutations, show significant enrichment in neurodevelopmental disorders (11,41)". This is not true. In Kaplanis et al 2020, damaging missense variants are robustly shown to contribute to NDDs (see SM figure 3 for example).

Response 2.6.3: Thank you very much for pointing out the fact that the damaging missense mutations contribute to the NDD. We have changed the sentence in the revised manuscript and included damaging missense in the sentence (Page 16, lines 21-23).

Comment 2.6.4: The data availability statement is weak. Many similar studies have deposited sequencing data from NDD cohorts to appropriate repositories.

Full Revision

Response 2.6.4: We agree with the reviewer's suggestion, however, due to the restrictions of ethical approval, we may not be able to deposit sequence data to public databases even with controlled access.

Comment 2.6.5: The authors should consider making the code used for their analysis open source, as this would help clarify some of the methodological questions I, and other may, have.

Response 2.6.5: We have made available code used to calculate the expected number of DNMs in a set of enhancers and cohort size on GitHub (<https://github.com/santoshatanur/expDNM>).

CROSS-CONSULTATION COMMENTS

I agree with the other reviews.

Reviewer #2 (Significance (Required)):

This is in important area of research, as the fraction of ID explained by non-coding variants is unknown. However, the very small sample size, especially when compared with other sequencing studies of NDDs in the literature, unfortunately limit the significance of the advance. Nevertheless, if authors can show that the results reported in the paper are robust, then the findings will be of interest to both researchers and clinicians studying NDDs.

My area of expertise is in the generation and analysis of sequencing data to study psychiatric and neurodevelopmental disorders. I have a lot of experience analysing exome sequencing data from proband-parent trios. I do not have experience with CRISPR, so I have not commented on that part of the study.

Reviewer #3 (Evidence, reproducibility and clarity (Required)):

Summary

In this manuscript, Vas and Boulet et al. presents the potential regulatory role of de novo mutations (DNMs) in intellectual disability (ID). They performed whole-genome sequencing in an ID cohort including 21 ID probands and their healthy parents. To study the regulatory DNMs in ID, they combined 17 ID probands without pathogenic coding DNMs with a previous cohort including 30 exome-negative ID cases. Leveraging their DNM dataset with a variety of epigenomic datasets, they observed ID DNMs were enriched more within fetal brain enhancers than adult brain enhancers. They also detected that the enhancers harboring ID DNMs showed promoter-enhancer interactions for the ID-relevant genes. Moreover, they identified recurrent mutations within enhancer clusters associated with CSMD1, OLFM1, and POU3F3 genes, when combining with larger pre-existing databases of genetic variants. Finally, they found that many ID DNMs were predicted to disrupt binding motifs of TFs, and experimentally validated the regulatory function of some of these loci. They showed the allele-specific activity for an enhancer region including an ID DNM for the SOX8 gene via luciferase assay as an episomal assay. They further showed that the same enhancer region regulates SOX8 expression by

performing CRISPRi, and proved the allele-specific impact of the same DNM via also genome editing with CRISPR/Cas9.

Major

Comment 3.1: The sample size of the Whole Genome Sequencing conducted in this study is extremely limited, and therefore the conclusions that can be drawn from the study are also extremely limited. The authors combined their data with existing cohorts for a subset of analyses, however, the novelty and utility of the findings from this cohort alone is limited.

Response 3.1: The fact that our sample size is small has been sufficiently addressed in the manuscript. However, we have applied robust statistical methods and used state of art experimental techniques to support our findings. Even with the smaller sample size, we show that the DNMs in ID patients are enriched in fetal brain enhancers as compared to adult brain enhancers. We identified three enhancer clusters with recurrent mutations and one of them was replicated in a large cohort. Because our sample size was small, we performed extensive experimental validations. We show that nine DNMs, from nine different ID patients, that are located in fetal brain enhancers show allele-specific expression. Furthermore, we show that *SOX8* enhancer DNM indeed affects *SOX8* expression using CRISPR knock-in. Though our sample size is small, with strong experimental support, we believe our findings are widely applicable.

Comment 3.2: Multiple testing burden must be considered when conducting enrichment studies in genomic regions using WGS data. Unfortunately, it is not considered here and without this the observed enrichment is not convincing. See for example <https://www.nature.com/articles/s41588-018-0107-y>.

Response 3.2: In the manuscript, we have presented the outputs of multiple independent analyses where we applied different statistical tests. In any analysis, if more than one hypothesis was tested, we applied multiple test correction. In the manuscript, we clearly mentioned whether the test is significant at a nominal p-value or after multiple test corrections. For example, enrichment analysis for developing cortex and prefrontal cortex cell types. Here we mention that “On the contrary, all four developed human brain cell types showed significant enrichment for ID DNMs compared to GoNL DNMs in promoter regions after correcting for multiple tests.” (Page 11, lines 12-14).

However, we agree that in the original manuscript we did not apply the multiple test correction for fetal vs adult brain enrichment analysis. In the revised manuscript, we have now applied multiple test corrections for fetal vs adult brain enrichment analysis. To achieve uniformity throughout the manuscript, we used R package `p.adjust` to estimate the false discovery rate (FDR) after multiple test corrections for all the analyses where more than one hypothesis test was performed.

- 1) DNM enrichment in fetal vs adult brain enhancers
- 2) Enrichment of known ID genes in the genes associated with the DNM-containing fetal brain enhancers
- 3) DNM enrichment analysis for developing brain and developed brain cell types
- 4) Recurrent DNMs in enhancer clusters

The gene ontology enrichment and tissue enrichment analysis for genes associated with the DNM-containing enhancers were performed using the web-based tool Enrichr (<https://maayanlab.cloud/Enrichr/>), which applies Bonferroni correction for all the tests. Similarly, tissue enrichment analysis for transcription factors whose binding sites were disrupted by the DNM was also performed using Enrichr. Hence for both of these analyses, p-values provided by Enrichr were reported in the manuscript.

The enrichment analysis of genes that are intolerant to loss of function mutations in genes associated with DNM-containing enhancers was a single test so multiple test correction was not applied.

In the revised manuscript, we have now applied multiple test correction to all the analyses where it was appropriate to apply. In the revised manuscript, we have now mentioned the statistical test used, the p-value obtained and the FDR for all the statistical tests.

Comment 3.3: The total number of promoter enhancer interactions as shown in Figure 2 is unbelievably high. The number of gene loops previously detected using Hi-C is much lower. This analysis seems to assign every enhancer in the region to the promoter within a TAD, which is much too liberal an analysis and not consistent with number of gene loops detected via Hi-C or eQTL work.

Response 3.3: As explained in detail in the manuscript to identify enhancer-promoter interactions, we used promoter capture Hi-C data and correlation of H3K27ac signal across 127 tissues/cell types available through a roadmap epigenomic project. On average each enhancer was associated with 1.64 genes and each gene was interacting with 4.83 enhancers. These findings were consistent with previous reports of enhancer-promoter interactions (25). We added this to the revised manuscript (Page 8, lines 25-27)

The specific genes presented in Figure 2 might have a higher number of enhancers associated with them because of the specific genomic architecture in those regions. For example, the TAD containing the *CSMD1* gene is a single gene TAD.

Comment 3.4: Because the total number of DNMs are few, I would recommend moving genomic annotations to hg38 rather than losing 123 DNMs via liftover to hg19.

Response 3.4: As we mentioned in the manuscript, we used a large amount of publicly available epigenomic datasets which are mostly available in hg19. To move the analysis to HG38 we need to liftover all the epigenomic datasets to HG38, which is much more complicated than liftover of DNMs to hg19.

Comment 3.5: The source of the neural progenitors used in the experiments are not described.

Response 3.5: We have differentiated hESC (H9) to NPCs, methods are now detailed in the manuscript under the heading “NPC culture protocol” (Page 29, lines 22-25).

Comment 3.6: The non-targeting or control gRNA is not described.

Response 3.6: Control gRNA is now described in the method (Page 30, line 7).

Comment 3.7: It's difficult to transfect both neural progenitors and neurons, it would be useful to see images of GFP expression if this is on the plasmid to know the degree of transfection efficiency and give greater confidence in the results presented in Figure 4.

Response 3.7: We agree it is difficult to transfect these cells, Hence we have transfected NPCs followed by a selection of transfected cells using antibiotics.. (detailed in the manuscript methods section Page 31, lines 6-7)

Comment 3.8: The specific instances where a one-tailed statistical test was used need to be highlighted.

Response 3.8: Apologies for the error, we used a two-tailed t-test throughout the manuscript. The method section is corrected accordingly.

Comment 3.9: At page 11, the authors stated "As enhancer regions of none of the human brain cell types showed significant enrichment for ID DNMs, we concentrated on DNMs overlapping enhancers from the bulk fetal brain for downstream analysis." However, cell-type-specific enhancer enrichment analysis vs fetal brain enhancer enrichment are two different analyses. The authors did not test if the ID DNMs were enriched more in fetal brain enhancers than control DNMs were. They only compared enrichment of ID DNMs and control DNMs fetal vs adult brain enhancers. Hence, this statement was not clearly justified. It would be improved by performing a fisher's exact test to assess if ID DNMs showed more enrichment within fetal bulk brain enhancers than control DNMs did similar to cell-type-specific enrichment analysis.

Response 3.9: Thank you very much for pointing out this. In the revised manuscript, we have removed the above-mentioned sentence from the manuscript.

Comment 3.10: At page 13, the authors indicated that "The fetal brain enhancer DNMs from ID probands frequently disturbed putative binding sites of TFs that were predominantly expressed in neuronal cells ($P = 0.022$; Table S12b). Our results suggest that the enhancer DNMs from ID probands were more likely to affect the binding sites of neuronal transcription factors and could influence the regulation of genes involved in nervous system development through this mechanism." How this conclusion is drawn is unclear. Table S12b includes three cell-types with identical p-values and odd ratios based on a statistical test. How could the authors get identical parameters for all cell-types? Which dataset was used to compare the expression of these transcription factors? Were transcription factors also expressed in non-neuronal cell-types? I would request the authors to clarify the analysis performed here in the methods section, and to compare the expression of TFs in other cell-types in order to conclude as "TFs that were predominantly expressed in neuronal cells". Also, this analysis would be improved by assessing the overlap of DNMs disturbed putative binding sites within cell-type-specific ATAC-seq peaks i.e. if they were enriched more within neuronal ATAC-seq peaks than non-neuronal ATAC-seq peaks.

Full Revision

Response 3.10: The results presented in the manuscript are the output of the tissue/cell type expression analysis performed using the web-based tool Enrichr (<http://amp.pharm.mssm.edu/Enrichr/>). In the method section of the original manuscript under the heading “Gene enrichment analysis”, we described that the “Gene ontology enrichment and tissue enrichment analysis were performed using the web-based tool Enricher (<http://amp.pharm.mssm.edu/Enrichr/>)”.

To estimate the tissue specificity of the gene expression Enricher uses gene expression data from the ARCHS4 project, which contains processed RNA-seq data from over 100,000 publicly available samples profiled by the two major deep sequencing platforms HiSeq 2000 and HiSeq 2500.

In supplementary table 12b of the original manuscript, we presented only cell types that showed significant enrichment. However, in the revised manuscript, we have provided a list of all the tissues and cell types tested by Enrichr along with corresponding p-values. Except for the neuronal cell types, none of the tissues and cell types showed statistically significant enrichment.

Furthermore, to make it clear we separated various gene and tissue enrichment analyses under different headings and provided a detailed explanation in the method section of the revised manuscript. The analysis of tissue specificity of transcription factor expression is now mentioned under the heading “**Enrichment of analysis for tissue/cell type expression of transcription factors whose binding site were affected by enhancer DNMs**” (Page 27, lines 10-17) and described it in the main text as well (Page 9, lines 14-17).

Comment 3.11: The authors randomly selected DNMs from 11 ID patients that were predicted to alter TFBS affinity for experimental validation in the luciferase assay. Were the allele-specific impacts of DNMs shown in Figure 3 consistent with the predicted impact via motifbreakR? Given that the authors prioritized the regulatory ID DNMs based on motifbreakR results for the experimental validation, I would request the authors to evaluate if the alleles disrupting a TF motif that mainly has activator/repressor function also showed lower/higher luciferase activity. That would help to support the evidence for the regulatory function of other ID DNMs predicted to be TF disruption but which could not be experimentally validated.

Response 3.11: Thank you very much for the excellent suggestion. We evaluated if the allele disrupting TF motif that mainly has activator/repressor function also showed lower/higher luciferase activity. It is more complex because of nine DNMs that showed allele-specific activity only five disrupt the TF motif and four of them result in the gain of the TF binding site.

Of the five that disrupt TF binding site, two disrupt the binding site of the activator (SP1 and CREB1) and both show reduced luciferase activity, while two disrupt the binding site of repressor or negative regulator (TCF7L1 and FOXN1) and both show increased luciferase activity. One DNM disrupts the binding site of the histone acetyltransferase (EP300) and shows reduced luciferase activity.

Of the four DNMs that result in a gain of transcription factor binding sites, two create a binding site for activator (HBP1 and BPTF) and show increased activity in luciferase activity. Of the two

gain of TFBS DNMs show reduced activity one creates TFBS for MAFB which can act as both a repressor and activator, while the second creates TFBS for HOXD13 for which we haven't found any support for the repressive activity. Taken together eight out of nine DNMs show increased or decreased luciferase activity, which matches the known role of TF whose binding site was disrupted or created by DNM.

In the revised manuscript, we added two additional columns in Table S13 indicating the role of the transcription factor (activator/repressor) and luciferase activity (gain or loss). Furthermore, we included the following text in the manuscript "Furthermore, for the majority of the DNMs (8 out of 9) the allele-specific activity was consistent with the predicted effect of the MotifBreakR (Table S13). For example, *CSMD1* enhancer DNMs disrupt the binding site of TCF7L1, a transcriptional repressor and luciferase assay shows that the mutant allele results in a gain of enhancer activity." (Page 14, lines 16-19).

Comment 3.12: At page 24 in the methods section, the authors defined the control DNMs set as "We downloaded de novo mutations identified in the healthy individuals in genomes of the Netherland (GoNL) study (21) from the GoNL website". Does DNM set from GoNL also include protein-truncating mutations? If it does, are there any de novo mutations that were previously also found in any other neurodevelopmental condition as being pathogenic or likely pathogenic? If it includes both protein-truncating de novo mutations and noncoding DNMs, the two datasets used for the analysis described in Figure 1 would not be appropriately comparable to conclude that regulatory DNMs in ID were enriched in fetal brain enhancers whereas control DNMs enriched in adult brain enhancers. In which enhancer category (fetal or adult) ID DNMs would be enriched if the same analysis is performed by using both protein-truncating and regulatory DNMs? I would request the authors to evaluate the possibility that regulatory DNMs were enriched more in fetal brain enhancers compared to adult brain regardless of disease status, if the GoNL control group includes both protein-truncating and regulatory DNMs. Also, as described in the previous statement, if control DNMs include only regulatory DNMs or both protein-truncating+regulatory DNMs is not clear. This analysis would also be improved by restricting control DNMs into regulatory DNMs.

Response 3.12: Of 11,020 GoNL DNMs, only six DNMs were protein-truncating. None of the six protein-truncating DNMs were reported to be pathogenic or likely pathogenic in clinvar for any of the neurodevelopmental disorders or any other disease. All 47 ID samples are coding negative means they don't have pathogenic or likely pathogenic coding DNM (protein truncating or damaging missense). Similarly, none of the GoNL samples has any pathogenic and likely pathogenic DNM. Hence, the comparison between ID cohort and GoNL cohort is a valid comparison.

However, as suggested by the reviewer, we performed multiple analyses. i) We performed enrichment analysis after removing six protein-truncating DNMs from GoNL cohort but the results did not change. ii) We excluded all protein-coding DNMs including synonymous and non-synonymous DNMs from both cohorts (included only non-coding DNMs) but the results did not change.

The number of DNMs that overlapped the fetal brain enhancer and adult brain enhancer did not change in any comparison. This is because protein-coding regions of the genome and, fetal and adult brain enhancers are mutually exclusive, they don't overlap. Therefore, the inclusion or exclusion of protein-truncating DNMs in enhancer enrichment analysis did not affect the results.

Comment 3.13: At page 14, the authors indicated that "In the heterozygous mutant clone, the *SOX8* gene showed a significant ($P = 0.0301$) reduction in expression levels, however, no difference was observed in expression levels of the *LFM1* gene ($P = 0.8641$; Fig. 4d), suggesting that the enhancer specifically regulates the *SOX8* gene but not the *LFM1* gene." based on the knock-in experiment for DNM. However, they did not show how CRISPRi of the enhancer which is the promoter for *LFM1* impacted on *LFM1* gene expression as they provided for the *SOX8* gene in Figure 4b. I would request the authors to rephrase the statement as "the regulatory impact of DNM within the enhancer is specific for *SOX8* but not for *LFM1*", or provide evidence that *LFM1* expression levels did not change after the CRISPRi experiment. Also, if the CRISPRi experiment would not show any change in *LFM1* expression, I would also request the authors to interpret what could be potential factors for that a regulatory sequence in a gene promoter would not impact its expression.

Response 3.13: As suggested by the reviewer, we have rephrased the sentence to "the regulatory impact of DNM within the enhancer is specific for *SOX8* but not for *LFM1*". (Page 15, lines 22-23)

We would like to point out that the **DNM-containing enhancer is not located in the promoter region of the *LMF1* gene**, but it is located downstream of the gene as *LMF1* is on the reverse strand. The genes *SOX8* (forward strand) and *LMF1* (reverse strand) share a promoter region as they are transcribed in the opposite direction. The DNM-containing enhancer that interacts with the promoter region of both *SOX8* and *LMF1* is located downstream of the *LMF1* gene. The region where gRNA was targeted for the CRISPRi experiment was approximately 10.5kb away from the 3' end of the *LMF1* gene.

Comment 3.14: The authors utilized neuroblastoma cells for luciferase assay, neuronal progenitor cells for CRISPRi, and HEK293T cells for genome editing CRISPR/Cas9 experiments. Given the cell-type-specificity of active regulatory elements, I would request the authors to provide more justification for the utilization of different cell types for each assay. More specifically, *LMF1* gene expression did not alter, albeit DNM's position in the gene promoter in Figure 4d. Could it be due to the low expression level of cell-type-specific transcription factors in HEK cells? Showing that expression levels of TFs whose binding motifs were disrupted via DNM at the region are comparable between HEK cells vs neuronal cells would be helpful here.

Response 3.14: We set out to perform the studies in neuroblastoma cells and validate the findings in NPCs. However, due to the difficulty in performing precise editing of a single nucleotide in neuroblastoma cells/NPCs, we have used HEK293T cells (Page 15, lines 12-14).

As described in the manuscript and the answer to the previous question, the DNM-containing enhancer is not located in the promoter region of the *LMF1* gene (promoter is near the *SOX8*

gene), but it is located downstream of the gene as *LMF1* is in the reverse strand of the genome. The region where gRNA was targeted for the CRISPRi experiment was approximately 10.5kb away from the 3' end of the *LMF1* gene, not in the promoter region of the *LMF1* gene.

Comment 3.15: Citation of many datasets are missing throughout the text including the (1) expression data in prefrontal cortex in the sentence at page 9 ".. but also predominantly expressed in the prefrontal cortex", (2) again expression data from neuronal datasets in the sentence at page 13 "The fetal brain enhancer DNMs from ID probands frequently disturbed putative binding sites of TFs that were predominantly expressed in neuronal cells", (3) NPCs in the sentence at page 14 "To investigate whether the putative enhancer of the SOX8/LMF1 gene indeed regulates the expression of the target genes, we performed CRISPR interference (CRISPRi), by guideRNA mediated recruitment of dCas9 fused with the four copies of sin3 interacting domain (SID4x) in the neuronal progenitor cells (NPCs).", and (4) H3K27ac and H3K4me1 datasets used in Figure 4e and described at page 14 in the sentence "Hence, we investigated H3K4me1 and H3K27ac levels at DNM containing SOX8 enhancer.". Adding citations of all external datasets utilized in the paper would be helpful for the reproducibility of the analyses and experiments.

Response 3.15: In the revised manuscript, we have included citations for datasets used in the analysis.

Analysis (1) and (2) were performed using the web-based tool Enrichr (<https://maayanlab.cloud/Enrichr/>). To perform tissue-specific expression analysis Enrichr uses the gene expression data from the ARCHS4 project (<https://maayanlab.cloud/archs4/>). We have mentioned this both in the text and the methods section of the revised manuscript.

(3) source of NPC is now mentioned in the methods section (Page 29, lines 22-25).

(4) The H3K4me1 and H3K27ac levels at DNM containing enhancers were measured using ChIP-qPCR in this study hence citation was not provided (Page 15, line 27 to page 16, line 1).

Comment 3.16: At page 10, the authors indicated that "We did not find enrichment for ID DNMs in open chromatin regions (ATAC-seq peaks) for any of the developing brain cell types" and on page 11, they stated, "On the contrary, all four developed human brain cell types showed significant enrichment for ID DNMs compared to GoNL DNMs in promoter regions after correcting for multiple tests". Given that ID DNMs were more enriched in fetal brain enhancers than adult brain enhancers in Figure 1, it is important to discuss why ID DNMs were enriched within developed brain cell-type regulatory elements but not in developing brain cell-type specific regulatory elements. I would request the authors to clarify this discrepancy. Could the distance to the gene be a factor in this discrepancy? How do cell-type-specific enrichment results change if ATAC-seq peaks from developing human cortex would be also restricted by chromatin accessibility regions within gene promoters (e.g. within +/- 2kb from TSS)? If ID DNMs within promoter regions were enriched within at least one of the cell-type-specific regulatory elements in both developing and adult brains, re-evaluating the analysis performed in Figure 1 by considering the distance of DNMs to genes would be critical to conclude temporal-specific enrichment of ID DNMs.

Response 3.16: ATAC-seq data from the developing brain was obtained from Song et al (2020, Nature) paper. ATAC-seq peaks open chromatin regions which include the entire regulatory

Full Revision

spectrum including active and inactive regulatory regions, therefore open chromatin regions may not show enrichment for DNMs.

To identify open chromatin regions that interact with the promoters that are active in specific cell types Song et al (2020, Nature) performed histone 3 lysine 4 trimethylation (H3K4me3) proximity ligation-assisted chromatin immunoprecipitation sequencing (PLAC-seq). Using cell type-specific chromatin interaction data, we investigated whether interacting open chromatin regions are enriched for ID DNMs as compared to GoNL DNMs. We found that interacting chromatin regions from IPCs were enriched for ID DNMs suggesting that DNMs affecting highly interacting regulatory regions might be functional.

Furthermore, as suggested by the reviewer we performed an enrichment analysis by restricting ATAC-seq peaks to +/-2kb region around the TSS of protein-coding genes. We found that ID DNMs were enriched in promoter regions of all four developing brain cell types. We have included this result in the revised manuscript (page 11, lines 5-8).

We then investigated if any of the 83 DNMs that overlapped with the fetal brain-specific enhancers or human gain enhancers were located within +/-2kb of the TSS of protein-coding gene. We found that only 4 DNMs were located within the 2kb region around TSS, suggesting that the enrichment observed fetal brain enhancers was not due to DNMs located in promoter regions.

Minor

Comment 3.17.1: In general, the study could benefit from more figures rather than providing results with tables to follow and understand them, especially for Table S6 and Table S11.

Response 3.17.1: The data from Table S6 is already represented in Figure 1 of the manuscript.

Comment 3.17.2: At figure 2, the colors of the arcs do not match the colors indicated in the label.

Response 3.17.2: We have changed the arc colours in the Figure 2 legends to reflect the real colours of the arc from “pink” to “magenta” and “green” to “dark green”.

Comment 3.17.3: At tables 11a and 11c, the column names indicated in the E and F columns are the same, it would be good to distinguish them.

Response 3.17.3: Thank you very much for pointing out the error. In table 11a and 11c of the revised manuscript, we have changed the column names of the E and F columns.

Comment 3.17.4: At page 10, the authors indicated that "The IPCs give rise to most neurons (32) hence DNMs in highly connected active promoters and enhancers from IPCs might have a profound impact on neurogenesis." This sentence is not clear.

Response 3.17.4: We have rephrased the sentence to make it clearer “suggesting that DNMs affecting highly interacting regulatory regions of IPCs might be functional” (Page 11, lines 3-4).

Full Revision

Comment 3.17.5: Radical glia -> radial glia

Response 3.17.5: We have changed it throughout the manuscript

Comment 3.17.6: Describe background gene lists used for all hypergeometric/fisher's exact tests.

Response 3.17.6: We have already mentioned the background gene list used for all hypergeometric/fisher's exact tests performed in the respective supplemental tables. For the analysis performed using the web-based tool Enrichr (<https://maayanlab.cloud/Enrichr/>), in the method section of the revised manuscript, we have mentioned the background gene set used by Enrichr to perform tissue enrichment analysis.

Comment 3.17.7: In Figure 4a, it would be useful to label the de novo mutation, otherwise it's not clear why a specific region was highlighted. Also, to highlight where the gRNA was targeted for the CRISPRi experiment.

Response 3.17.7: In Figure 4a, we have labelled the de novo mutation in the revised manuscript. We have added panel 4b to highlight the region where gRNA was targeted for the CRISPRi experiment.

CROSS-CONSULTATION COMMENTS

I agree with the other reviewers' comments. I just have one specific comment: Reviewer 1 suggested that RNA-seq would be more accurate than gene expression; however, I feel that this assay is not necessary and may be quite expensive for the targeted gene expression differences measured here.

Reviewer #3 (Significance (Required)):

Overall this study attempted to identify and validate novel non-coding variants associated with ID. However, given limitations in sample size, statistical testing, and experimental design, as described above, many of these conclusions are limited.

January 13, 2023

Re: Life Science Alliance manuscript #LSA-2022-01843

Dr. Santosh S Atanur
Imperial College London
Department of Metabolism, Digestion and Reproduction
Hammersmith Hospital Campus
Du Cane Road
London W12 0NN
United Kingdom [GB]

Dear Dr. Atanur,

Thank you for submitting your revised manuscript entitled "Regulatory de novo mutations underlying intellectual disability" to Life Science Alliance. The manuscript has been seen by the original reviewers whose comments are appended below. While reviewer 1 continues to be overall positive about the work in terms of its suitability for Life Science Alliance, Reviewer 2 raises few important issues.

Our general policy is that papers are considered through only one revision cycle; however, given that the suggested changes are relatively minor, we are open to one additional short round of revision. Please note that I will expect to make a final decision without additional reviewer input upon resubmission.

Please submit the final revision within one month, along with a letter that includes a point by point response to the remaining reviewer comments.

To upload the revised version of your manuscript, please log in to your account: <https://lsa.msubmit.net/cgi-bin/main.plex>
You will be guided to complete the submission of your revised manuscript and to fill in all necessary information.

B. MANUSCRIPT ORGANIZATION AND FORMATTING:

Sincerely,

Reviewer #1 (Comments to the Authors (Required)):

The authors addressed my previous concerns.

Reviewer #2 (Comments to the Authors (Required)):

I thank the authors for clarifying my questions, as a result the methods used to evaluate enrichment of DNMs in fetal enhancers is now much more clearly described and understood. However, I still have major concerns about whether their approach to test for DNM enrichment, which involves normalising observed and expected DNMs in adult brain enhancers to the background mutation rate for fetal brain enhancers, is appropriate. I am not aware of any other studies using this method, and as the P values presented appear hugely inflated given the very small sample, this suggests that the statistics are not well calibrated. If other studies have used this method, it would be useful to cite them. If this is indeed a novel method, I would expect the authors to present additional data supporting the robustness of their approach. I would be helpful if the authors could test their method with a negative control, to ensure the reader that the statistics are well calibrated. For example, does the method produce non-significant results when comparing DNV rates across enhancers that are unlikely to have a role in ID?

Moreover, in the methods section the authors write "Because of the difference in background mutation rate, we cannot directly compare the number of DNMs between fetal brain enhancers and angular gyrus.". I don't think this is true. The authors can use a two-sample Poisson rate ratio test to compare DNMs between these sets. As an example in R, if the observed and expected DNMs in Brain Anular gyrus is 22 and 31.97, respectively, and in fetal brain specific enhancers 53 and 31, respectively, then a two sample Poisson test would be `poisson.test(c(53,22),c(45.62,31.97))`, which gives a P of 0.045 and a rate ratio of 1.68. This test supports their conclusions that DNMs are enriched in fetal enhancers, but with much weaker evidence that is likely a true reflection of the study's power.

Finally, the observed number of mutations in the GoNL cohort are in general much lower than the expected number. This suggests that the mutation rates being used are not calibrated for this data set. Have the authors considered coverage in their mutation rate models (for both the GoNL cohort and their ID cohorts)? They should adjust their models for low coverage, which is standard practice.

Reviewer #1

Comment 1.1: The authors addressed my previous concerns.

Response 1.1: We thank the reviewer for finding our response satisfactory.

Reviewer #2

Comment 2.1: I thank the authors for clarifying my questions, as a result the methods used to evaluate enrichment of DNMs in fetal enhancers is now much more clearly described and understood. However, I still have major concerns about whether their approach to test for DNM enrichment, which involves normalising observed and expected DNMs in adult brain enhancers to the background mutation rate for fetal brain enhancers, is appropriate. I am not aware of any other studies using this method, and as the P values presented appear hugely inflated given the very small sample, this suggests that the statistics are not well calibrated. If other studies have used this method, it would be useful to cite them. If this is indeed a novel method, I would expect the authors to present additional data supporting the robustness of their approach. I would be helpful if the authors could test their method with a negative control, to ensure the reader that the statistics are well calibrated. For example, does the method produce non-significant results when comparing DNV rates across enhancers that are unlikely to have a role in ID?

Response 2.1: The method used in the manuscript to estimate the enrichment of DNMs is new and has not been used by any other study. However, we believe when comparing enrichment between enhancer sets of two tissues it is important that both enhancer sets be normalised to the same background mutation rate for valid comparison.

As suggested by the reviewer, to ensure our statistics are well calibrated we used enhancer sets of two tissues (lung and small intestine) that may not have any role in ID and for which enhancer data was available for both fetal and adult stages in Roadmap Epigenomics project. We did not find enrichment of DNMs in fetal enhancers compared to adult enhancers from the lung and small intestine ($P=0.45$). This result suggests that our statistics are well-calibrated, and our finding is a true reflection of biology rather than a technical artefact. The result of this analysis is presented in the revised manuscript on page 8, lines 6-11.

Comment 2.2: Moreover, in the methods section the authors write "Because of the difference in background mutation rate, we cannot directly compare the number of DNMs between fetal brain enhancers and angular gyrus.". I don't think this is true. The authors can use a two-sample Poisson rate ratio test to compare DNMs between these sets. As an example in R, if the observed and expected DNMs in Brain Anular gyrus is 22 and 31.97, respectively, and in fetal brain specific enhancers 53 and 31, respectively, then a two sample Poisson test would be `poisson.test(c(53,22),c(45.62,31.97))`, which gives a P of 0.045 and a rate ratio of 1.68. This test supports their conclusions that DNMs are enriched in fetal enhancers, but with much weaker evidence that is likely a true reflection of the study's power.

Response 2.2: We thank the reviewer for pointing to the statistical test that can be used to compare two enhancer sets whose background mutation rate is different. We aggregated enhancer sets of all adult brain subsections to get one enhancer set for the adult brain. Similarly, we combined fetal and human gain enhancers set to get one enhancer set for the fetal brain. A total of 82 DNMs overlapped with fetal brain enhancers and 99 DNMs overlapped with adult brain enhancers. The expected number of DNMs for fetal brain enhancers was 70.97 DNMs while 131.05 DNMs for adult brain enhancers. We observed significant enrichment of DNMs in fetal brain enhancers as compared to adult brain enhancers ($P=0.005$) when we performed a two-sample Poisson rate ratio test. However, we did not observe an enrichment of DNMs in our negative controls, fetal lung and fetal small intestine, compared to adult lung and adult small intestine ($P=0.58$ and 0.88 respectively). This suggests that the results obtained using our method are robust and our conclusions hold true irrespective of the statistical test applied. These results are presented in the revised manuscript on page 8, lines 17-25.

Comment 2.3: Finally, the observed number of mutations in the GoNL cohort are in general much lower than the expected number. This suggests that the mutation rates being used are not calibrated for this data set. Have the authors considered coverage in their mutation rate models (for both the GoNL cohort and their ID cohorts)? They should adjust their models for low coverage, which is standard practice.

Response 2.3: We are aware and agree with the reviewer that coverage is different between our study and the GoNL. In the GoNL study sequencing was performed on Illumina HiSeq 2000 with 13X coverage while samples in our study were sequenced on the Illumina HiSeq X10 platform with an average coverage of 37X. Hence, in our mutation model, we adjusted for the coverage, however, as the comparison was performed within a cohort (ID or GoNL) between fetal and adult brain enhancers and not between two cohorts, the results did not change.

February 8, 2023

RE: Life Science Alliance Manuscript #LSA-2022-01843R

Dr. Santosh S Atanur
Imperial College London
Department of Metabolism, Digestion and Reproduction
Hammersmith Hospital Campus
Du Cane Road
London W12 0NN
United Kingdom

Dear Dr. Atanur,

Thank you for submitting your revised manuscript entitled "Regulatory de novo mutations underlying intellectual disability". We would be happy to publish your paper in Life Science Alliance pending final revisions necessary to meet our formatting guidelines.

-please add the legend for your supplementary figure to the main manuscript text

A. FINAL FILES:

B. MANUSCRIPT ORGANIZATION AND FORMATTING:

****It is Life Science Alliance policy that if requested, original data images must be made available to the editors. Failure to provide**

original images upon request will result in unavoidable delays in publication. Please ensure that you have access to all original data images prior to final submission.**

The license to publish form must be signed before your manuscript can be sent to production. A link to the electronic license to publish form will be sent to the corresponding author only. Please take a moment to check your funder requirements.

Sincerely,

February 13, 2023

RE: Life Science Alliance Manuscript #LSA-2022-01843RR

Dr. Santosh S Atanur
Imperial College London
Department of Metabolism, Digestion and Reproduction
Hammersmith Hospital Campus
Du Cane Road
London W12 0NN
United Kingdom

Dear Dr. Atanur,

Thank you for submitting your Research Article entitled "Regulatory de novo mutations underlying intellectual disability". It is a pleasure to let you know that your manuscript is now accepted for publication in Life Science Alliance. Congratulations on this interesting work.

DISTRIBUTION OF MATERIALS:

Again, congratulations on a very nice paper. I hope you found the review process to be constructive and are pleased with how the manuscript was handled editorially. We look forward to future exciting submissions from your lab.

Sincerely,
